# Physiology regulates the relationship between coccosphere geometry and growth-phase in coccolithophores

Rosie M. Sheward[1,2], Alex J. Poulton[3], Samantha J. Gibbs[1], Chris J. Daniels[3], Paul R. Bown[4]

[1] Ocean and Earth Science, University of Southampton, National Oceanography Centre, Southampton, SO14 3ZH, United Kingdom.
[2] Institute of Geosciences, Goethe-University Frankfurt, 60438 Frankfurt am Main, Germany.
[3] Ocean Biogeochemistry and Ecosystems, National Oceanography Centre, Southampton, SO14 3ZH, United Kingdom.
[4] Department of Earth Sciences, University College London, Gower Street, London, WC1E 6BT, United Kingdom.

*Correspondence to*: Rosie M. Sheward (sheward@em.uni-frankfurt.de)

**Abstract.** Coccolithophores are an abundant phytoplankton group that exhibit remarkable diversity in their biology, ecology, and calcitic exoskeletons (coccospheres). Their extensive fossil record is testament to their important biogeochemical role and is a valuable archive of biotic responses to environmental change stretching back over 200 million years. However, to realise the full potential of this archive for (paleo-)biology and biogeochemistry requires an understanding of the physiological processes that underpin coccosphere architecture. Using culturing experiments on four modern coccolithophore species (*Calcidiscus leptoporus*, *Calcidiscus quadriperforatus*, *Helicosphaera carteri* and *Coccolithus braarudii*) from three long-lived families, we investigate how coccosphere architecture responds to shifts from exponential (rapid cell division) to stationary (slowed cell division) growth phases as cell physiology reacts to nutrient depletion. These experiments reveal statistical differences in coccosphere size and the number of coccoliths per cell between these two growth phases, specifically that cells in exponential-phase growth are typically smaller with fewer coccoliths, whereas cells experiencing growth-limiting nutrient depletion have larger coccosphere sizes and greater numbers of coccoliths per cell. Although the exact numbers are species-specific, these growth-phase shifts in coccosphere geometry demonstrate that the core physiological responses of cells to nutrient depletion result in increased coccosphere sizes and coccoliths per cell across four different coccolithophore families (Calcidiscaceae, Coccolithaceae, Isochrysidaceae, Helicosphaeraceae), a representative diversity of this phytoplankton group. Building on this, the direct comparison of coccosphere geometries in modern and fossil coccolithophores enables a proxy for growth phase to be developed that can be used to investigate growth responses to environmental change throughout their long evolutionary history. Our data also show that changes in growth rate and coccoliths per cell associated with growth-phase shifts can substantially alter cellular calcite production. Coccosphere geometry is therefore a valuable tool for accessing growth information in the fossil record, providing unprecedented insights into the response of species to environmental change and the potential biogeochemical consequences.

# 1 Introduction

The fossil remains of biomineralised plankton provide comprehensive records of their biogeography, ecology, diversity and evolution that have significance for our understanding of past ocean and climate systems and its influence on these microscopic organisms. Despite their small size (2 to 200 μm for nanno- and microplankton), the vast numbers of photosynthesising plankton in the ocean drive many regional- to global-scale biogeochemical processes and comprise the biomass that sustains the ocean ecosystem (e.g., Menden-Deuer and Kiørboe, 2016). Investigating the biological response of plankton species to environmental variability is therefore a crucial step in understanding the potential consequences of future climate change on marine systems.

Coccolithophores are a major group of calcifying marine algae that first evolved more than 200 million years ago (Ma) during the Late Triassic (Janofske, 1992; Bown et al., 2004). The remains of their calcite cell coverings contribute to the export of biogenic carbonate to deep-sea sediments (Broecker and Clark, 2009), forming a geographically and temporally extensive fossil record that is mostly in the form of individual calcite plates called coccoliths. Spatial and temporal analysis of coccoliths reveals the evolution, biogeography and ecology of past species (e.g., Haq and Lohmann, 1976; Knappertsbusch, 2000; Ziveri et al., 2004; Gibbs et al., 2006; Baumann et al., 2016) and the response of species and communities to palaeoceanographic and palaeoclimatic variability (e.g., Bollmann et al., 2002; Bown, 2005; Bollmann et al., 2009; Bown and Pearson, 2009).

Valuable new insights into past coccolithophore communities can also be provided by the study of intact fossil coccospheres that have not disarticulated into their component coccoliths, providing intriguing snapshots of individual cell growth in geological time (Gibbs et al., 2013; Bown et al., 2014; O'Dea et al., 2014). Whilst the preservation of intact coccospheres in sediments is generally uncommon, recent investigations showcase a large diversity of coccospheres from a range of ages, ocean basins, and latitudes in numbers suitable for robust quantitative analysis (Gibbs et al., 2013; Bown et al., 2014). The discovery of relatively abundant fossil coccospheres in exceptionally well-preserved sedimentary deposits inspired Gibbs et al. (2013) to first explore the quantitative links between coccosphere geometry (coccosphere size, coccolith length and coccolith number) and population growth. Their laboratory experiments using the modern species *Coccolithus braarudii* and *Emiliania huxleyi* identified that cells undergoing rapid cell division (termed 'exponential-phase' growth) were smaller and had fewer coccoliths per coccosphere compared to cells dividing slowly, or not at all ('stationary-phase' growth). This initial evidence for a relationship between growth phase and coccosphere geometry was then used to reconstruct the response of fossil taxa (*Coccolithus* and *Toweius*) through an interval of rapid warming ~56 Ma called the Paleocene-Eocene Thermal Maximum (Gibbs et al., 2013; O'Dea et al., 2014). As growth phases describe 'states' of rapid or slowed growth rates, these findings hint that coccosphere geometry could provide opportunities for new insights into the ecological 'fitness' and subsequent evolutionary success of coccolithophore populations where growth rates (or other physiological measures) cannot be measured directly.

The development of coccosphere geometry as an indicator, or even proxy, of growth phase in the fossil record requires further evidence that phases of rapid and slowed growth produce quantifiably distinct differences in coccosphere geometry, which can be regarded as a 'universal' feature of coccolithophores rather than just a species-specific attribute. Even across the diversity of modern species, we observe substantial variability in cell size, coccolith length and numbers of coccoliths per cell. Given this observation, can we reasonably hypothesise that the growth-geometry relationship reported by Gibbs et al. (2013) for two modern species is similar across coccolithophores in general? If this is the case, then coccosphere geometry could prove to be a valuable proxy for growth phase and provide new insights into important fitness-related traits where growth rates cannot be measured directly. One potential concern is that coccolithophores show pronounced species-specific and even strain-specific physiological responses to a variety of environmental manipulations such as carbonate chemistry and nutrient availability in culture experiments (Langer et al., 2006; Langer et al., 2009; Krug et al., 2011), which may extend to coccosphere variability. We therefore require coccosphere geometry data from multiple modern species experiencing different growth phases in order to further investigate the relationship between coccosphere geometry and growth.

Here, we aim to determine relationships (if any) between growth phase and coccosphere geometry in three modern coccolithophore species - *Calcidiscus leptoporus*, *Calcidiscus quadriperforatus* and *Helicosphaera carteri* - and to integrate these new data with those previously determined by Gibbs et al. (2013) for *Coccolithus* and *Emiliania*. *Calcidiscus* and *Helicosphaera* are particularly pertinent study taxa, as they have widespread modern and geological occurrences and are important components of mid- to low-latitude coccolithophore communities, preferring warmer temperate to tropical waters (Ziveri et al., 2004). These species are also three of the largest and most heavily calcified of all the modern species, along with *Coccolithus pelagicus* in the high latitudes and *Coccolithus braarudii* in the mid- to high-latitudes (Ziveri et al., 2004). They are therefore important contributors to the production (Daniels et al., 2014; Daniels et al., 2016) and export of inorganic carbon to the deep ocean (Ziveri et al., 2007). Variability in coccosphere geometry in these species, particularly the number of coccoliths per cell, could therefore substantially alter cellular calcite with significant consequences for calcite production and export. The well-documented fossil records of these genera extend back to the first occurrence of *Calcidiscus* ~57 Ma (Bown et al., 2007) and *Helicosphaera* ~54 Ma (Perch-Nielsen, 1985). Alongside *Coccolithus* they have been significant components of coccolithophore communities over much of the last ~55 Ma (Perch-Nielsen, 1985; Bown et al., 2007).

*Helicosphaera* and *Calcidiscus* also have distinct evolutionary and physiological differences that may highlight restriction of the growth-geometry relationship to specific lineages. Species within the Helicosphaeraceae (Order Zygodiscales) have evolved in a lineage quite separate to the Coccolithaceae and Calcidiscaceae (Order Coccolithales), with the two orders diverging very early in coccolithophore evolutionary history during the Jurassic, ~150-200 Ma (de Vargas et al., 2007). *Helicosphaera carteri* is also physiologically distinct from both *Coccolithus* and *Calcidiscus* species as it is motile in the diploid (heterococcolith-bearing) life-cycle phase. As *Coccolithus* and *Calcidiscus* are considered to be relatively closely related, though still classified into separate families, we would predict that the growth-phase diagnostic features of

coccosphere geometry in *Calcidiscus* species might be more like those reported for *Coccolithus* by Gibbs et al. (2013). To our knowledge, the experiments undertaken for this study have produced the most extensive dataset of modern coccosphere

geometry yet to be presented, comprising a total of more than 13,300 measurements of coccosphere and cell size, coccolith length and coccoliths per cell from 2,850 individual cells.

## 2 Methods

### 2.1 Experiment design

Monoclonal cultures of South Atlantic Ocean *Calcidiscus quadriperforatus* strain RCC 1135, *Calcidiscus leptoporus* strain

RCC 1130 and *Helicosphaera carteri* strain RCC 1323 were obtained from the Roscoff Culture Collection (RCC) and maintained at an incubation temperature of 19 °C at the National Oceanography Centre, Southampton. Cultures were acclimated to new experimental temperature and light conditions for a minimum of two weeks (>10 generations) prior to the start of each experiment. The light regime remained consistent across all experiments at irradiance levels of 75 to 90 µmol photons $m^{-2}$ $s^{-1}$ (equivalent to a daily photon flux of ~3.5 mol photons $m^{-2}$ $d^{-1}$) with a 12-hour light, 12-hour dark irradiance

cycle. To achieve a range of cell division rates, experiments were undertaken at 16, 18, 20 and 22 °C, which is well within the natural temperature range experienced by field populations of these three species (Ziveri et al., 2004).

        For each temperature experiment, all three species were cultured simultaneously and in duplicate following a 'batch culture' procedure, where an initially low number of cells $ml^{-1}$ are left to increase in density, using up nutrients, until initial nutrient levels are completely depleted and population growth ceases. This approach enables coccosphere geometry data to

be collected from both nutrient-replete rapid cell-division days and nutrient-deplete slowed cell-division days towards the end of the experiment, as used successfully in the experiments of Gibbs et al. (2013) for *Coccolithus*. The initial starting density of cells for each experiment was ~300 cells $ml^{-1}$ (taken from acclimated cultures) added to 350 ml of sterilised and filtered natural seawater enriched with 28.8 µM nitrate and 1.8 µM phosphate (lower-nutrient K/20 medium, modified from Keller et al., 1987, following Langer et al., 2006 and Daniels et al., 2014). The effect of increasing cell density on the

carbonate chemistry of the media over the duration of the experiment was not directly quantified but it is likely that there was DIC consumption throughout the course of each experiment. However, our aim was to minimise the effect of cell growth on carbonate chemistry by: using low-nutrient media to ensure that cultures reached nutrient-limiting conditions relatively quickly and at relatively low final cell concentrations; using 650 ml polycarbonate flasks (Thermo Fisher Scientific) with vented lids to allow faster diffusive gas exchange between the culture media and the atmosphere outside the

flask; and aerating and mixing each flask daily under sterile conditions to further encourage gas exchange. After initial inoculation of the media, experiment cultures increase in cell number rapidly, termed the exponential growth phase, and were allowed to grow into stationary phase, at which point increasing nutrient limitation reduces growth rates such that the day-to-day increase in cells $ml^{-1}$ decreases towards zero. The typical experiment duration between initial inoculation and the onset of stationary phase growth was between 14 and 21 days.

## 2.2 Growth rate calculation

Daily cell abundance was determined from triplicate counts of cells ml$^{-1}$ using a Sedgwick Rafter Cell (Pyser-SGI; following Langer et al., 2006) on a transmitted light microscope at x100 magnification. As *H. carteri* is a motile species, 40 µl per ml (4% final volume) 10% formaldehyde was added to *H. carteri* samples prior to counting to inhibit movement and ensure counting accuracy. Daily growth rates were calculated as the natural log of the difference in cell density between the census day and the day before (Langer et al., 2006). The duration of the exponential growth phase was then determined by visual examination of these daily growth rates and plots of cell abundance over time. Mean exponential growth rates ($\mu$) for each temperature experiment were calculated from daily cell abundances, where $\mu = [\ln(N_1) - \ln(N_0)]/d$, and $N_0$ and $N_1$ are the cell concentrations at the beginning and end of the exponential phase, respectively, and d is the duration of the exponential phase in days.

## 2.3 Coccosphere geometry

Samples for light microscope (LM) analysis were taken daily using 2-5 ml of each culture replicate, filtered onto cellulose nitrate filters (pore size 0.8 µm; Sartorius Stedim Biotech) and dried overnight at 50 °C. One half of each filter was then fixed between a glass microscope slide and a cover slip using Norland Optical Adhesive 74 (Norland Products Inc.) and cured under UV light exposure. All LM analysis was performed using a cross-polarised light microscope (Olympus BX51) with a colour camera attached (Olympus DP71). Coccosphere geometry data was obtained through LM following the same techniques applied by Gibbs et al. (2013) and Daniels et al. (2014), and described in detail here. Random transects across the widest section of the filter hemisphere were performed until 30 individual coccospheres per slide were located from slides corresponding to alternate day or, in some instances, daily samples. First, the number of coccoliths around each cell ($C_N$) was counted by finely adjusting focal depth. Then, in-focus images of the upper coccosphere surface and maximum cell cross-section were captured from which biometric measurements (Fig. 1) of coccolith length ($C_L$), coccosphere size (Ø; size including calcite covering) and cell size (Θ; size excluding calcite covering and which is assumed to be equivalent to cell diameter) were taken (Cell^D software, Olympus). Unlike the spherical coccospheres of *Coccolithus* and *Calcidiscus* species, *H. carteri* coccospheres are prolate spheroids (Fig. 4), so here we report cell and coccosphere sizes for this species as equivalent spherical diameters. Prolate spheroid volume is calculated as V= ($\pi/6$)$d^2$h, where d is the short-axis cell/coccosphere diameter and h is the long-axis cell/coccosphere height (Sun and Liu, 2003). This volume is used to calculate equivalent spherical radius. This coccosphere geometry dataset is available from https://doi.pangaea.de/doi:10.1594/PANGAEA.865403.

## 2.4 Cellular calcite calculation

Particulate inorganic carbon (PIC) per cell was calculated for each individual coccosphere following Young and Ziveri (2000):

$$Cellular\ PIC\ (pmol\ C\ cell^{-1}) = \frac{C_N \times C_L{}^3 \times k_s \times 2.7}{100} \tag{1}$$

where $C_N$ is number of coccoliths per cell, $C_L$ is coccolith length (µm), $k_s$ is a shape factor that numerically describes species-specific coccolith morphology, and 2.7 pg µm$^{-3}$ is the density of calcite. Division by 100 calculates cellular PIC in pmol C cell$^{-1}$ from pg cell$^{-1}$. We use the shape factors of $k_s = 0.08$ for *Calcidiscus* spp., $k_s = 0.05$ for *H. carteri*, and $k_s = 0.06$ for *Coccolithus* spp. from Young and Ziveri (2000). Mean, 25th and 75th percentiles, and the range of cellular PIC were calculated from the 22 °C experiment data of each species using coccosphere geometry data from selected mid-exponential-phase days (*C. leptoporus* = days 7, 9, 11; *C. quadriperforatus* = days 3, 5, 7; *H. carteri* = days 6, 7, 8) and all non-exponential-phase days. Mean exponential- and non-exponential-phase calcite production rates at 22 °C were calculated based on these mean cellular calcite values multiplied by mean exponential and non-exponential growth rates, respectively, for the same temperature experiment. The minimum to maximum range in growth rates was based on growth rates observed across all temperature experiments.

## 2.5 Additional experimental results from *Coccolithus*

This study reports the new experimental results for *Calcidiscus* and *Helicosphaera* alongside coccosphere geometry and growth data for *Coccolithus* from two previous studies that used identical LM methods to collect coccosphere geometry data. Gibbs et al. (2013) obtained coccosphere geometry data from a comparable batch culture experiment at a single temperature in *Coccolithus braarudii* strain RCC 1197. This data is presented for direct comparison with the three new species of this study, as much of the Gibbs et al. (2013) data was originally presented as Supplementary Information to accompany that short-format paper. We also present results from a previously unanalysed dataset of exponential-phase coccosphere geometry in *C. braarudii* strain RCC 1198 and *C. pelagicus* strain RCC 4092, originally published as a data report by Sheward et al. (2014) and available from http://www.pangaea.de (doi: 10.1594/PANGAEA.836841). For that study, batch culture experiments were undertaken at multiple temperatures (6-12 °C in *C. pelagicus* and 12-19 °C in *C. braarudii*) and samples for coccosphere geometry analysis collected on a single mid-exponential-phase experiment day (further details in Daniels et al., 2014).

## 2.6 Statistical analyses

The relationships of Ø with $C_L$ and $C_N$ in each species were tested by Model II reduced major axis (RMA) linear regression analysis. Confidence intervals (95 %) for the regression slope were calculated by bootstrapping over 1999 iterations using the freeware Paleontological Statistics (PAST; v. 3.13; Hammer et al., 2001). We compare species-specific mean Ø and mean $C_N$ between growth phases using a *t*-test in GraphPad Prism (version 7.0a for Mac OS X; GraphPad Software, Inc., USA). The difference in mean Ø or $C_N$ between exponential-phase growth and non-exponential-phase growth were considered significant at $p < 0.05$.

## 3. Results

### 3.1 Growth rates

The four temperature experiments resulted in a modest range of mean exponential growth rates ($\mu$) across *Helicosphaera* and *Calcidiscus* species. The highest mean exponential growth rate for *C. quadriperforatus* was achieved at 22 °C ($\mu = 0.44$ d$^{-1}$), for *C. leptoporus* at 20 °C ($\mu = 0.44$ d$^{-1}$), and for *H. carteri* at 20 °C ($\mu = 0.45$ d$^{-1}$). Mean exponential growth rates for *C. braarudii* at 15 °C were 0.68 d$^{-1}$. These values are well within the ranges reported in other studies carried out at similar temperatures for *Calcidiscus* (Langer et al., 2006; Buitenhuis et al., 2008; Fiorini et al., 2010; 2011; Langer et al., 2012; Candelier et al., 2013; Müller et al., 2014) and *H. carteri* (Stoll et al., 2002; Šupraha et al., 2015). Exponential growth rates of 0.4-0.5 d$^{-1}$ signify that roughly half of the culture population undergoes cell division each day. Maximum cell density was ~100,000 cells ml$^{-1}$ in *C. leptoporus* cultures, 60-100,000 cells ml$^{-1}$ for *C. quadriperforatus*, ~30,000 cells ml$^{-1}$ for *H. carteri* and ~25,000 cells ml$^{-1}$ for *C. braarudii*.

### 3.2 Within-species range in coccosphere geometry

Coccosphere ($\varnothing$) and cell size ($\Theta$), coccolith length ($C_L$) and number of coccoliths per cell ($C_N$) show clear species-specific differences (Fig. 2, Table 1). A considerable range in $\varnothing$ is seen in all species; 13.8 to 24.4 μm in *C. quadriperforatus*, 9.4 to 20.9 μm in *H. carteri* and 10.0 to 19.7 μm in *C. leptoporus*. This is a comparable $\varnothing$ range to *C. pelagicus* (11.7 to 20.8 μm) but slightly less than the $\varnothing$ range observed in *C. braarudii* (13.7 to 29.7 μm). Cell size exhibited a similarly large range of 6.5 to 18.0 μm in *H. carteri*, 6.4 to 16.5 μm in *C. leptoporus*, 8.6 to 18.8 μm in *C. quadriperforatus*, 7.9 to 18.1 μm in *C. pelagicus*, and 9.9 to 15.8 μm in *C. braarudii* (Table 1).

*Calcidiscus* spp. and *H. carteri* show a much greater range in $C_N$ compared to *Coccolithus* spp. (Fig. 2e-h). The most frequently observed $C_N$ is 16 in *H. carteri* cells, 18 in *C. quadriperforatus* cells, and 19 in *C. leptoporus* cells, with a maximum number of ~30 coccoliths in all of these species. In one *C. leptoporus* cell, the coccosphere was formed from 45 coccoliths (Fig. 4c). In contrast, *Coccolithus* cells more typically have 11 to 14 coccoliths per cell, up to a maximum of 20 coccoliths. The relationship between $C_N$ and $\varnothing$ subsequently shows a steeper gradient in *Helicosphaera* and *Calcidiscus* (greater $C_N$ increase per μm $\varnothing$) compared to *Coccolithus* (Fig. 2). The similar coccosphere sizes but significantly greater number of coccoliths per coccosphere of *C. quadriperforatus* compared to *C. braarudii*, and *C. leptoporus* compared to *C. pelagicus*, indicates that *Calcidiscus* species achieve a greater degree of coccolith overlapping compared with *Coccolithus* species of a similar coccolith size. This is likely the result of the circular shape and narrower central tube structure in *Calcidiscus* coccoliths, which therefore pack more tightly around the cell with increasing $C_N$, moderating a corresponding increase in $\varnothing$. The minimum $C_N$ in *H. carteri* is similar to *Coccolithus* ($C_N = 6$ and $C_N = 5\text{-}7$, respectively). The smallest *H. carteri* cells, with just 6 coccoliths, formed cuboid coccospheres (Fig. 4a) and are most likely recently-divided cells. Cubiform coccospheres have also been reported in Bown et al. (2014) for the extinct Paleogene taxa *Toweius pertusus* and

*Umbilicosphaera bramlettei* and 'boxy' coccospheres are also seen in several *Chiasmolithus* species, which are probably also related to small cell sizes soon after cell division.

Although coccosphere geometry is similar in the two *Calcidiscus* species (Fig. 2f, g), it is not identical, with *C.*
225 *leptoporus* producing coccospheres with a slightly greater $C_N$ on average than *C. quadriperforatus* (slopes of 3.01 and 2.11, respectively). In contrast, the two species of *Coccolithus* are more closely comparable, with the linear regression gradient between Ø and $C_N$ 1.50 and 1.35 in *C. pelagicus* and *C. braarudii*, although the gradients are offset from each other (y-intercepts of -10.17 and -17.33, respectively; Fig. 2h). Until recently, these two *Calcidiscus* species were considered to be intraspecific morphotypes (Knappertsbusch et al., 1997; Knappertsbusch, 2000) or sub-species (Geisen et al., 2002) but have
230 since been shown to be genetically-distinct, which is also the case for *C. pelagicus* and *C. braarudii* (Sáez et al., 2003; de Vargas et al., 2004). The considerable overlap in $C_L$, Ø and $C_N$ in *Calcidiscus* species makes species-differentiation based solely on any one of these parameters difficult. However, the species-specific coccosphere geometry identified here lends further support to the genetic distinction between these species, alongside previously identified morphological and ecological differences (Knappertsbusch et al., 1997; Knappertsbusch, 2000; Geisen et al., 2002; Renaud et al., 2002; Sáez et al., 2003;
Geisen et al., 2004; Baumann et al., 2016).

Coccolith length varies between cells by up to 4.5 μm in *H. carteri*, 6.0 μm in *C. quadriperforatus*, and 3.7 μm in *C. leptoporus*, which is similar to $C_L$ ranges of 3.0 to 8.5 μm reported in selected studies on sediment samples (e.g., Baumann, 2004; Henderiks and Törner, 2006; Herrmann et al., 2012; Baumann et al., 2016). Unfortunately, no culturing experiments on *Calcidiscus* or *Helicosphaera* report $C_L$ measurements for comparison. In contrast to $C_N$, $C_L$ shows no relationship with Ø
within these clonal populations (Fig. 2i-l) and superimposing $C_N$ onto plots of Ø against $C_L$ (Fig. 2m-p) clearly demonstrates the strong co-variance of Ø and $C_N$. In our clonal populations, cells have relatively restricted ranges in Ø and $C_L$ that have no statistically-significant relationship (Fig. 2i-l). A weak relationship between Ø and $C_L$ appears to exist in *Coccolithus* when data for *C. pelagicus* is combined with data from two strains of *C. braarudii* (Fig. 2l, p). This $C_L$-Ø relationship only occurs in these culture experiments when data from several growth-synchronised populations are mixed. This effect is also seen in
the culture and field data of Gibbs et al. (2013) and is greatly amplified in fossil assemblages, which typically integrate the remains of surface populations over longer time-spans (Gibbs et al., 2013, their Fig. 3a). In our single-clone culture populations, however, the principle coccosphere geometry relationship is between $C_N$ and Ø rather than $C_L$ and Ø.

### 3.3 Coccosphere geometry as a function of growth

This study demonstrates that coccosphere size in all the species studied is statistically smaller during days of rapid, nutrient-
250 replete, exponential-phase growth than during days of slowed, nutrient-depleted, non-exponential-phase growth (Fig. 3). Mean Ø across all four temperature experiments during exponential-phase growth is 14.8 μm in *H. carteri*, 18.4 μm in *C. quadriperforatus*, 13.1 μm in *C. leptoporus* and 20.5 μm in *C. braarudii*. Mean coccosphere diameter during non-exponential growth is modestly but statistically (unpaired t-test) larger than during exponential-phase growth, with mean Ø 0.55 μm larger in *C. quadriperforatus* (t=3.324, df=839, p<0.001), 0.64 μm larger in *H. carteri* (t=4.659, df=990, p<0.0001),

and 0.90 μm larger in *C. leptoporus* (t=5.669, df=1020, p<0.0001). Mean Ø in *C. braarudii* (Gibbs et al., 2013) shows a larger increase of 1.34 μm (t=9.216, df=548, p<0.0001) between exponential- and non-exponential-phase growth. An increase in cell size has also previously been observed in response to nutrient limitation in *Coccolithus* and *Helicosphaera* (Gerecht et al., 2014; Gerecht et al., 2015; Šupraha et al., 2015).

In addition to size differences, coccospheres also typically consist of fewer coccoliths during exponential-phase growth and a greater number of coccoliths during non-exponential-phase growth (Fig. 3). This is shown by an increased frequency of cells in higher $C_N$ classes and an increased mean $C_N$ during non-exponential phase growth in each species. Cells no longer able to maintain exponential rates of growth have an average of 1 to 2 extra coccoliths per cell in *H. carteri* (t=5.067, df=990, p<0.0001) and *C. quadriperforatus* (t=5.451, df=840, p<0.0001), 2 to 3 extra coccoliths per cell in *C. leptoporus* (t=6.312, df=1020, p<0.0001) and 3 to 4 extra coccoliths per cell in *C. braarudii* (t=14.24, df=548, p<0.0001). The frequency distribution of $C_N$ for each species (Fig. 3) can be used as a quantitative indicator of whether cells are in a recently-divided state (close to the minimum number of coccoliths per cell observed, $C_N \leq 10^{th}$ percentile of the data) or are in a ready-to-divide state (close to the maximum number of coccoliths per cell observed, $C_N \geq 90^{th}$ percentile of the data). These $C_N$ 'thresholds' for recently-divided and ready-to-divide cells for each species are shown in Fig. 3 and Table 1. Based on the species-specific geometries observed, recently-divided cells typically have $C_N \leq 12$ in *H. carteri* and $C_N \leq 14$ in *Calcidiscus* spp., whilst cells that are ready to divide have $C_N \geq 21$ in *H. carteri*, $C_N \geq 23$ in *C. quadriperforatus*, and $C_N \geq 25$ in *C. leptoporus* (Fig. 3). During exponential growth, the mean and frequency distribution of population $C_N$ is skewed towards the minimum observed $C_N$ and therefore the population has a higher percentage of 'recently-divided' coccosphere geometries. Populations exhibiting slowed growth are more likely to have an increased percentage of cells in a 'ready-to-divide' state. However, there are always some 'recently-divided' cells and some 'ready-to-divide' cells in both exponential- and non-exponential-phase populations due to on-going cell division, albeit at different rates. There is therefore a large overlap in Ø and $C_N$ size-range between exponential- and non-exponential-phase populations (Fig. 3), with negligible change in the maximum Ø and $C_N$ of each (Table S1).

**3.4 Cellular particulate inorganic carbon**

PIC can be calculated directly from the extensive dataset of coccosphere geometry collated for this study by multiplying $C_N$ by individual coccolith calcite, following Eq. (1) (Sect. 2.4; Young and Ziveri, 2000). Mean exponential-phase cellular PIC (calculated at mid-exponential-phase for each temperature experiment) was 10.7 to 12.6 pmol C cell$^{-1}$ in *C. leptoporus*, 21.3 to 25.8 pmol C cell$^{-1}$ in *H. carteri*, but higher in *C. quadriperforatus* (21.5-30.0 pmol C cell$^{-1}$) and *C. braarudii* at (27.9 pmol C cell$^{-1}$) (Table 1). At 22 °C, mean PIC during non-exponential experiment days was 9 to 45 % higher compared to mid-exponential-phase across all species due to an increase in median $C_N$ of 2 to 4 coccoliths (Table S1). The 25$^{th}$ and 75$^{th}$ percentiles are also clearly shifted towards higher cellular PIC in cells no longer growing exponentially (Fig. 5a.). The 25$^{th}$ percentile increases 50 to 60% in *Calcidiscus*, with *C. braarudii* and *H. carteri* increasing by 20 to 25%. The increase in 75$^{th}$

percentile is not as large, but is still considerable in *C. leptoporus* and *C. braarudii* at 36% and 24%, respectively, with *C. quadriperforatus* and *H. carteri* showing more modest increases of 6% and 11%.

## 4. Discussion

### 4.1 Physiological insights into coccosphere geometry

Within these experiments, coccosphere size ($\emptyset$) and the number of coccoliths per cell ($C_N$) varied depending on whether the culture population was increasing in cell numbers each day at a rapid rate (exponential-growth phase) or a slowed rate (non-exponential-growth phase). Across all four species investigated, the transition from exponential into non-exponential phase growth was clearly associated with a shift towards cells with a greater $C_N$ (mean $C_N$ increased by 1-3 coccoliths per cell) and larger coccosphere sizes (mean $\emptyset$ increased by 0.6 μm in *H. carteri* and *C. quadriperforatus*, 0.9 μm in *C. leptoporus* and 1.3 μm in *C. braarudii*; Fig. 3). This represents a significant increase of 4 to 7% on exponential-phase mean $\emptyset$ and an increase of 10 to 27% on exponential-phase mean $C_N$ (*t*-test, p<0.0001). $C_N$ is not a frequently recorded variable but where $\emptyset$ and $C_N$ in both nutrient-replete and nutrient-deplete cultures can be inferred from supplementary information (Balch et al., 1993; Paasche, 1998; Gerecht et al., 2014; Gerecht et al., 2015; Šupraha et al., 2015) these are consistent with the extensive observations from our experiments for *Calcidiscus* and *H. carteri* and those of Gibbs et al. (2013) for *C. braarudii*.

The relationship between growth phase, $\emptyset$ and $C_N$ can be understood by considering the process of cell division and how it is affected by the nutrient depletion that instigates non-exponential-phase growth. Both $\emptyset$ and $C_N$ vary as each cell progresses through the cell division cycle (unpublished observations; Taylor et al., 2007; Müller et al., 2008). Recently-divided cells are small with approximately the minimum number of coccoliths required to form a complete cell covering (unpublished observations; Fig. 4). After division, cells recommence coccolith production and increase $C_N$ until the cell has sufficient coccoliths to cover two newly divided cells. Coccosphere diameter correspondingly increases alongside increasing $C_N$ as the cell synthesises organic cellular components such as proteins, lipids and carbohydrates. Cultures that are able to maintain exponential rates of cell division subsequently have a lower mean $\emptyset$, $\Theta$ and $C_N$ as the majority of cells are in a 'recently divided' state (Fig. 3, 4). When cells are no longer able to maintain exponential rates of cell division, in this instance due to decreasing nutrient availability, they divide less frequently on average. This is observed in the later days of each experiment as an increase in the mean $\emptyset$, $\Theta$ and $C_N$, an interpretation that is consistent with the findings of Gibbs et al. (2013).

An increase in cell size, $\Theta$, under decreasing nutrient availability may seem counterintuitive, as nutrients are essential for phytoplankton growth. Nitrate and phosphate are the two key nutrients required by most phytoplankton (Arrigo, 2005; Moore et al., 2013) and they fulfil different purposes within the cell. Phosphate limitation primarily impedes production of the RNA, phosolipids and DNA that are essential for cell replication, and phosphate is a key component of cellular energy carriers (Zhao et al., 2015). Nitrate limitation particularly impacts the synthesis of proteins and pigments used in photosynthesis (Zhao et al., 2015). However, despite the suppression of cell division and photosynthetic activity by

phosphate and nitrate limitation, respectively, the cell is still able to synthesise non-essential lipids and carbohydrates. Cell size and particulate organic carbon content (POC) are therefore able to increase under nutrient limited conditions (e.g., Müller et al., 2008). A similar increase in POC could also reflect DIC limitation, which sometimes results from DIC drawdown as cell numbers rise to high concentrations in non-exponential-phase growth. An increase in POC under DIC limitation was previously shown in *C. braarudii* (Rickaby et al., 2010).

The greater $C_N$ of coccospheres during non-exponential-phase growth (Fig. 3) includes the occurrence of some large coccospheres with very high $C_N$ (Fig. 4) and more than enough coccoliths to cover two daughter cells. This is evidence that cellular calcification (coccolith production) can proceed uninterrupted despite decreasing nutrient availability and indicates that the calcification process has a lower nutrient 'cost' compared to cell division processes (Paasche, 1998; Monteiro et al., 2016). This is also illustrated by the dramatic overproduction of coccoliths in *E. huxleyi* under nutrient limitation (Balch et al., 1993; Paasche, 1998), and supported by the $C_N$ evidence from *Calcidiscus* and *Helicosphaera* in this study and *Coccolithus* in Gibbs et al., (2013). An alternative possibility is that the continued production of coccoliths by cells in stationary phase leaves them poised and 'ready-to-divide' should nutrients become newly available. In support of this, the recommencement of cell division in stationary phase cultures after the addition of nutrient-replete seawater has been observed in *E. huxleyi* cultures (J. Young, pers. comm.).

### 4.2 Contrasting growth phase and growth rate

The clear relationship we observe between growth *phase*, Ø and $C_N$ is interpreted to be the result of cellular physiology (calcification, biomass production and the synthesis of molecules involved in cell division) responding to shifts in nutrient availability over the course of the experiments, with stationary-phase nutrient depletion decreasing growth rates to zero once levels became inhibiting to cell division. Exponential-phase growth *rates*, the proportion of the culture undergoing cell division between two consecutive days (daily growth rates) or averaged across multiple days (mean exponential growth rates), are instead affected by temperature (which determines the rate of nutrient uptake and the rate of metabolic cell processes), and irradiance (which affects photosynthetic rates, i.e., the rate at which the cell can produce energy). Our manipulation of experiment temperature (16-22 °C) aimed to achieve a range of exponential-phase growth rates that might reveal any correlation between growth *rate* and coccosphere geometry. However, no clear relationship between Ø, Θ, $C_L$ or $C_N$ and exponential growth rate (daily or mean) was observed in our experiments. One explanation for this might be that mean exponential growth rates ($\mu_{exp}$) were not sensitive enough to the temperature range we applied (*C. quadriperforatus* $\mu_{exp}$=0.30-0.44 $d^{-1}$; *C. leptoporus* $\mu_{exp}$=0.31-0.44 $d^{-1}$; *H. carteri* $\mu_{exp}$=0.28-0.45 $d^{-1}$). In addition, growth rates would not necessarily be expected to influence coccosphere geometry in the same way as a shift in growth phase caused by nutrient depletion, as temperature and light primarily affect physiological rates (e.g., Eppley, 1972; Falkowski et al., 1985), whilst nutrient limitation primarily impedes molecule synthesis (e.g., Zhao et al., 2015). Calcification, for example, is contingent on both the rate at which nutrients can be supplied to the cell (temperature-dependent) and processed into energy (light-dependent) and can proceed under nutrient limitation, as shown in our experiments, but may be less efficient under sub-

optimal temperature or light conditions. As yet, no studies have investigated the response of cell size and/or coccosphere geometry under a range of optimum *vs.* limiting temperature or light conditions in coccolithophores. As growth phase describes two different physiological states, one of which manifests as slowed to zero daily growth rates caused by depleted nutrient availability, the ability to identify coccolithophore populations with coccosphere geometries characteristic of each growth phase is an important advance in interpreting growth information directly from the coccosphere. Valuable additional perspectives on the specific role of growth rate on coccosphere geometry would be gained from future work using semi-continuous or continuous culturing techniques to achieve a range of steady-state exponential growth rates under different nutrient, temperature or light conditions.

### 4.3 Coccosphere geometry as a proxy for growth phase in the fossil record

A notable finding of this study is that coccosphere geometry (coccosphere size, coccolith length and coccoliths per cell) is species-specific but Ø and $C_N$ respond identically to growth phase changes across four different species of *Calcidiscus, Coccolithus* and *Helicosphaera*. This strongly suggests that coccosphere geometry within the major coccolithophore families Calcidiscaceae and Helicosphaeraceae responds to nutrient-driven changes in growth phase, and therefore cell physiology, in the same way as species within the families Coccolithaceae (Gibbs et al., 2013) and Noelaerhabdaceae (Balch et al., 1993; Paasche, 1998; Gibbs et al., 2013). This is compelling evidence that, as a group, coccolithophores express a common physiological response to shifts from exponential to non-exponential (stationary) growth phase, seen as a modest but significant increase in the average $C_N$ and Ø of a population (Fig. 3, 4). This specifically results from the ability of the cell to maintain calcification processes even when rates of cell division are suppressed by nutrient limitation.

One of the aims of this study was to further develop the proxy application of fossil coccosphere geometry first proposed by Gibbs et al. (2013) for *Coccolithus* and *Toweius*. Culture experiments on *Coccolithus* and *Emiliania huxleyi* showed that Ø and $C_N$ responded to growth phase as described above and Gibbs et al. (2013) applied this to coccosphere records of fossil *Coccolithus* and *Toweius* (an ancestor of *E. huxleyi*) across the Paleocene-Eocene Thermal Maximum climate change event (56 Ma). Given the tendency of coccolithophores to show strong species- and strain-specific responses to external factors, extending this application to other fossil species might be seen as highly speculative based on data from only two modern species. The new experimental data presented here for *Calcidiscus* and *Helicosphaera*, in combination with previous results for *Coccolithus* and *Emiliania* (Balch et al., 1993; Paasche, 1998; Gibbs et al., 2013; Gerecht et al., 2014; Gerecht et al., 2015), provides validity that coccosphere geometry persistently responds to growth phase in a common manner, regardless of species, and notably that mean population $C_N$ increases under slowed growth in *Calcidiscus*, *Helicosphaera*, *Coccolithus* and *Emiliania*.

To further develop this proxy we need to establish threshold values of $C_N$ that distinguish recently-divided cells and cells with theoretically sufficient coccoliths to undergo cell division. As an exponential-growth phase population is undergoing cell division at a rapid rate, it has a greater percentage of recently-divided cells ($C_N \leq$ lower threshold: Table 1). In contrast, a slowly-dividing population in stationary-phase growth has a greater percentage of ready-to-divide cells ($C_N \geq$

upper threshold; Table 1) but fewer recently-divided cells. Here, we report these $C_N$ threshold values (Table 1) for *Calcidiscus* and *Helicosphaera* (Fig. 3) and add them to those identified by Gibbs et al. (2013) for *Coccolithus*. $C_N$ is relatively easy to measure in both fossil and modern coccospheres using light microscopy, and so potentially provides a robust method for identifying populations that are growing rapidly (exponential populations where >~15% population is characterised by cells with $C_N$ typical of recently-divided cells) compared to populations that are growing slowly (non-

exponential populations where >~15% population is characterised by cells with $C_N$ typical of ready-to-divide cells). This is illustrated in Fig. 6 and these specific $C_N$ threshold values can be used to approximate the growth state of any fossil or modern population of *Coccolithus*, *Helicosphaera* or *Calcidiscus* species. In reality, the mixing of populations of different growth states in the fossil record (and open ocean) will frequently result in percentages of recently-divided and ready-to-divide cells lying between the two end-members shown in Fig. 6 (Gibbs et al., 2013). However, where time-series of

coccosphere geometry data are available, intervals of changing growth states can be identified as substantial temporal shifts in the proportional percentage of recently-divided to ready-to-divide cells indicative of less or more favourable growth conditions (Gibbs et al., 2013; O'Dea et al., 2014). Whilst in these experiments non-exponential growth phase is initiated by nutrient depletion, this would be an overly simplistic interpretation for modern field, sediment trap or fossil populations. It is more reasonable to interpret shifts in population coccosphere geometry as a response to less or more favourable growth

environments, incorporating a combination of nutrient, temperature, light and other environmental factors that may influence population growth.

        For fossil taxa that have no direct modern counterpart, the general characteristics of rapidly growing populations consisting of an increased proportion of smaller cells with fewer coccoliths relative to slowly dividing populations can be used as a qualitative indicator of changes in growth-phase through time. Based on the species studied here, the typical $C_N$ of

recently-divided and ready-to-divide cells can be tentatively proposed for any species (living or extinct) based on the 10[th] and 90[th] percentiles of the $C_N$ histogram produced from a compiled taxon-specific dataset of coccosphere geometry. Relative changes in the distribution of $C_N$ through time within any species can then provide a valuable indication of intervals where species may be experiencing nutrient conditions that are more (shift towards lower $C_N$, recently-divided geometry) or less (shift towards higher $C_N$, ready-to-divide geometry) favourable for growth. This can be achieved for any taxon by first

compiling a dataset of coccosphere geometry for the focal species and then calculating the 10[th] and 90[th] percentiles to estimate taxon-specific $C_N$ thresholds. Growth phase can then be estimated by calculating the percentage of each sample with coccospheres of ready-to-divide and recently-divided $C_N$ before plotting as Fig. 6. We would caution users to be mindful that the full range of coccosphere sizes in a species may not be represented in any particular sample and that, as fossil species are typically morphospecies concepts, the range in $C_N$ observed is likely to incorporate multiple intraspecies morphotypes or

ecotypes with similar but subtly varied coccosphere geometries. We therefore recommend that as many samples as is feasible are considered in the full dataset before calculating $C_N$ thresholds and that the minimum to maximum range in coccosphere geometry parameters (Table 1) within modern species are heeded as an indication of how variable coccosphere geometry can be within even a single clone. The coccolithophore fossil record is also vulnerable to size-related

preservational biases (Young et al., 2005) that may affect the abundance of very small or very large coccospheres within a sample and may not be consistent through time at the same site. The overall quality of preservation should therefore be considered when interpreting changes in coccosphere geometry through time, and caution should be exercised if there is a suspected strong bias against the preservation of particular taxa, very small coccoliths or very small or very large coccospheres within a sample.

Thus far, relatively common fossil coccospheres have been documented from at least 24 localities (Burns, 1975; Covington, 1985; Lambert, 1987; Young and Bown, 1991; Mai, 1997; Mai et al., 1998; Henderiks, 2008; Ciurej, 2010; Bown et al., 2014) representing low to high latitudes, the North and South Atlantic Oceans, North Pacific Ocean, Indian Ocean and Southern Ocean, and ranging from Kimmeridgian (Late Jurassic) to Pleistocene. Coccosphere geometry analysis is therefore likely to prove applicable at a range of localities and time intervals, but reasoned selection of sampling sections is likely to be important for retrieving sufficient coccospheres for robust data analysis. Hemipelagic sediments, particularly those with less intense bioturbation, are perhaps more likely to contain coccospheres than deep-sea oozes (Bown et al., 2014).

Whilst we conclude that coccosphere geometry can be used with confidence as a proxy of growth phase in the fossil record or modern ocean, we must be clear that the environmental and growth signal recorded in field populations is always more complex than any laboratory experiment result. Populations may only experience a specific nutrient state for a few weeks or less before conditions change and the coccosphere geometry response of any individual cell is likely to be further complicated by temperature and light conditions that are also essential for growth. At present there is little to no experimental data to demonstrate the response of coccosphere geometry to temperature or irradiance, or how changes in growth rate specifically (rather than growth phase) may manifest in coccosphere geometry. The fossil record of coccolithophores further compounds these considerations, as fossil assemblages are typically temporal integrations of many thousands of very short-lived population states. The coccosphere geometry signal of species populations transitioning between rapid and slowed growth phases clearly becomes obscured and diluted by the mixing of population remains and subtle shifts in species morphotypes and ecotypes as environmental conditions vary, as illustrated by Gibbs et al. (2013).

Nevertheless, the use of coccosphere geometry as a proxy for growth phase is valid across coccolithophores generally, and not just specific species. As exponential- and stationary-growth phases describe two distinct physiological states, the former with rapid growth rates experiencing optimal nutrient supply and the latter with slowed growth rates suffering nutrient limitation, coccosphere geometry provides a unique link to the physiology of individual cells, and may contribute towards our understanding of population fitness (measured as growth rate or fitness-related traits such as cell size and calcification) and ultimately the long-term success of species responding to varying nutrient conditions. As such, the coccosphere geometry-growth phase proxy is a highly valuable tool, for the first time allowing direct considerations of growth phase in evolutionary and palaeoceanographic studies.

**4.4 Implications of growth-driven cellular PIC and POC for calcite production**

*Coccolithus*, *Calcidiscus* and *Helicosphaera* are potentially major regional calcite producers in both the modern (Daniels et al., 2014; Daniels et al., 2016) and past ocean (Ziveri et al., 2007), as they are some of the largest, most heavily calcified modern species with distributions throughout sub-Polar (*C. pelagicus*), temperate (*C. braarudii*), and sub-tropical (*Calcidiscus* and *Helicosphaera*) oceans (Ziveri et al., 2004). The process of biogenic calcification is thought to be responsive to climate and particularly sensitive to changes in ocean carbonate chemistry (for reviews see Riebesell and Tortell, 2011; Bach et al., 2015; Meyer and Riebesell, 2015). Our experiments show that calcite per cell can also change significantly with growth phase as the number of coccoliths in the coccosphere varies in response to changing nutrient availability. This was not previously known for any species other than *E. huxleyi*, which produces high $C_N$, multi-layered coccospheres under nutrient limitation. *E. huxleyi* additionally sheds excess coccoliths into the surrounding waters (e.g., Balch et al., 1993), potentially amplifying the biogeochemical impact of increased coccolith production under low nutrient conditions, although, to our knowledge, this species is unique in this respect. Calcite production is a function of cellular calcite (particulate inorganic carbon, PIC) and growth rate, and could therefore change considerably with environmental conditions through time with implications for the biogeochemical cycling of carbon in the ocean.

PIC can be calculated directly from coccosphere geometry by multiplying $C_N$ by individual coccolith calcite, following Eq. (1) (Sect. 2.4; Young and Ziveri, 2000). Using the 22 °C experiment as an example, mean exponential-phase cellular calcite ranged from 10.91 pmol C cell$^{-1}$ in *C. leptoporus* to 19.3 pmol C cell$^{-1}$ in *C. quadriperforatus*. In the non-exponential-phase, cellular calcite increased by 4.9 pmol C cell$^{-1}$ (45%) in *C. leptoporus*, 2.6 pmol C cell$^{-1}$ (9%) in *C. quadriperforatus*, 2.9 pmol C cell$^{-1}$ (11%) in *H. carteri* and 5.9 pmol C cell$^{-1}$ (21%) in *C. braarudii* (Fig. 5; Table S1) due to the higher $C_N$ proportion of each population with greater $C_N$ (Fig. 3). Calcite production per cell per day (pmol C cell$^{-1}$ d$^{-1}$) can be calculated by multiplying cellular calcite (pmol C cell$^{-1}$) by growth rate (d$^{-1}$) (e.g., Daniels et al., 2014; 2016). Calcite production in these four species is 6 to 20 times higher than in *E. huxleyi* at a comparable growth rate (Fig. 5; Balch et al., 1996; Poulton et al., 2010), and hence these heavily calcified species (e.g., the calcite of one *C. braarudii* cell is equivalent to ~78 cells of *E. huxleyi*) do not necessarily need to be abundant or maintain comparative growth rates to still dominate calcite production (Daniels et al., 2014; Daniels et al., 2016). A dramatic difference in calcite production can be seen between populations growing exponentially and those no longer growing exponentially, with reductions in calcite production of 77 to 88% in all species due to the order of magnitude decrease in growth rates (based on mean exponential and non-exponential growth rates for the 22 °C experiment; Fig. 5c). In field populations, growth rates can reach as low as <0.2 d$^{-1}$ (Poulton et al., 2014), similar to the culture populations in slowed growth shown in Fig. 5, and therefore these shifts to such low calcite production per cell per day are approximate minimum calcite production values for these species. However, it is clear that rates of calcite production can be altered by up to 50% for even a moderate change of growth rate of 0.1 to 0.2 d$^{-1}$, for example where coccolithophore populations experience changes in nutrient supply, temperature or light availability that no longer support optimal rates of cell division (Poulton et al., 2010; 2014).

The majority of studies attribute environmentally-driven changes in calcite production to variation in calcite per coccolith, through coccolith size, thickness or malformation (e.g., Beaufort et al., 2011; Horigome et al., 2014). However, $C_L$ would need to increase by roughly 5 to 20% to achieve the same change in cellular calcite as that produced by an increase of just 2 to 4 coccoliths per cell based on our data. O'Dea et al. (2014) similarly found that changes in coccolith calcite mass of ~5 to 11% and ~6 to 16% for *Toweius pertusus* and *Coccolithus pelagicus* during the PETM were dwarfed by up to 500% changes in cellular calcite resulting from combined changes in $C_L$, Ø, and $C_N$ across the same time interval of Paleogene climate change. Change in $C_N$ with growth phase is therefore key when considering the impact of environmental parameters such as nutrient availability on cellular PIC and calcite production rates. The dominant control of growth rates on calcite production, as demonstrated recently by Gerecht et al. (2015) for *C. pelagicus*, is an important consideration that is often overlooked when investigating the impact of climate on long-term calcite production, carbon export, and sequestration and should be accounted for alongside growth phase changes in calcite.

## 5. Conclusions

Experiments on modern species of the coccolithophores *Calcidiscus* and *Helicosphaera* have shown significant differences in coccosphere geometry under exponential-phase growth (nutrient replete conditions) and non-exponential-phase growth (nutrient depleted conditions) identical to those previously observed in *Coccolithus* and *Emiliania huxleyi*. The extension of these earlier findings into two additional families demonstrates that the decoupling of cell division and calcification rates in coccolithophores is a core physiological response to nutrient depletion and is expressed in coccosphere geometry as an increase in coccoliths per cell and coccosphere size. With due consideration, coccosphere geometry can be applied as a proxy for growth phase in the geological record, as well as in sediment trap and modern field population samples, with the expectation that populations of any coccolithophore species experiencing growth-limiting nutrient conditions will have a greater number of larger cells with more coccoliths per cell. The variability of coccosphere geometry with growth, specifically calcite production through the production of coccoliths, identifies coccoliths per cell as an equally important parameter as calcite per coccolith in determining cellular calcite. Growth rate is the principal driver of calcite production rather than cellular calcite, highlighting the need for consideration of growth information in both the modern ocean and geological record in order to explore the impact of future climate change scenarios on calcite production and export.

**Data availability**

The coccosphere geometry data and accompanying culture conditions generated for this study are publically accessible as Sheward et al. (2016) at https://doi.pangaea.de/10.1594/PANGAEA.865403.

**Author contributions**

RS, AP and SG conceived the design of the experiment with advice from CD. RS performed the culturing experiments, collected and analysed the data. RS, SG and AP interpreted the findings. RS wrote the manuscript with contributions from all authors.

The authors declare that they have no conflict of interest.

**Acknowledgements**

We would like to thank Ian Probert (Roscoff Culture Collection) for his provision of the coccolithophore cultures used in this study and Lucie Munns for laboratory assistance. We thank Ros Rickaby for her useful discussions on a previous version of this manuscript. RS was funded by a Vice Chancellors Studentship from the University of Southampton and a Natural Environmental Research Council (NERC) award (reference 1272561). We acknowledge research support from NERC (AP NERC National Capability funding and CD via a NERC studentship) and the Royal Society (SG).

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

## Tables

**Table 1.** Summary statistics of species-specific coccosphere geometry data, PIC (particulate inorganic carbon), and $C_N$ (number of coccoliths per cell) thresholds for classifying the proportion of recently- and ready-to-divide cells in a population, based on the complete coccosphere geometry dataset from all experiment days. Summary statistics for both growth phases are shown in Table S1. The full dataset of experimental conditions, daily growth rates and coccosphere geometry measurements from each individual coccosphere is available as Sheward et al. (2016) at Pangaea.de, https://doi.pangaea.de/doi:10.1594/PANGAEA.865403.

| Parameter | | *Helicosphaera carteri* | *Calcidiscus leptoporus* | *Calcidiscus quadriperforatus* | *Coccolithus pelagicus* | *Coccolithus braarudii* |
|---|---|---|---|---|---|---|
| Number of values | | 990 | 1020 | 840 | 180 | 880 |
| Coccosphere diameter, Ø μm | Min Mean Max | 9.35 15.01 20.90 | 10.02 13.28 19.72 | 13.84 18.56 24.39 | 11.74 16.12 20.80 | 13.66 20.49 29.68 |
| Cell diameter, Θ μm | Min Mean Max | 6.53 12.01 17.99 | 6.39 9.90 16.54 | 8.64 13.72 18.81 | 7.94 12.89 18.11 | 9.92 16.36 25.83 |
| Coccolith length, $C_L$ μm | Min Mean Max | 6.70 8.89 11.22 | 5.02 6.72 8.76 | 5.67 9.10 11.67 | 5.68 8.95 11.59 | 7.87 12.21 17.32 |
| Coccoliths per cell, $C_N$ μm | Min Mean Max | 6 16 30 | 10 19 45 | 8 18 29 | 7 14 23 | 5 12 20 |
| PIC pmol C cell$^{-1}$ | Min Mean Max | 7.79 24.48 54.95 | 3.27 13.00 36.58 | 6.09 30.69 80.19 | 4.15 17.28 42.08 | 6.97 35.89 147.80 |
| Recently divided cells $C_N \le$ | | $C_N \le 12$ | $C_N \le 14$ | $C_N \le 14$ | $C_N \le 11$ | $C_N \le 8$ |
| Ready to divide cells $C_N \ge$ | | $C_N \ge 21$ | $C_N \ge 23$ | $C_N \ge 25$ | $C_N \ge 18$ | $C_N \ge 16$ |
| Reference | | This study | This study | This study | Sheward et al. (2014) | Gibbs et al. (2013); Sheward et al. (2014) |

## Figures

(a) Coccolith-focused

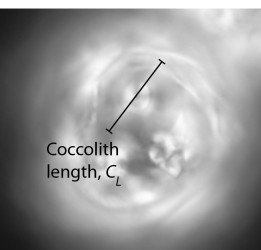

(b) Cross-section focused

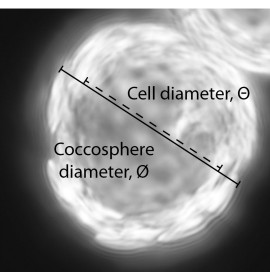

**Figure 1.** Light microscope image of a *C. quadriperforatus* coccosphere illustrating the coccosphere geometry terminology used in this study and the size measurements made on each individual coccosphere. After counting the number of coccoliths per cell ($C_N$), images are taken of (a) an in-focus, representative coccolith on either the top or bottom surface of the coccosphere from which coccolith length ($C_L$) is measured, and (b) a cross-sectional view from which the coccosphere diameter (Ø) and internal coccosphere diameter, assumed to represent cell diameter ($\Theta$), are measured.

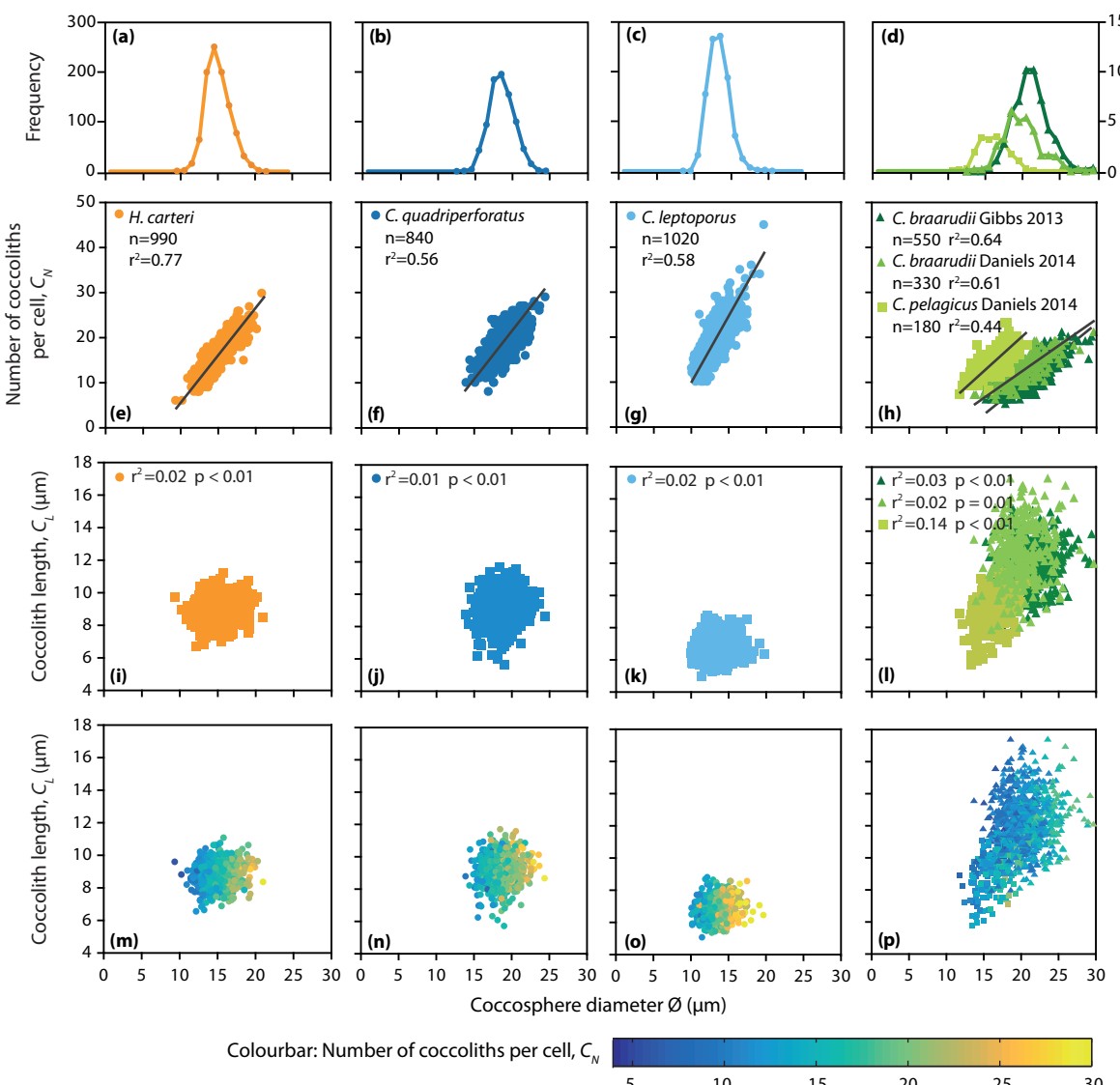

**Figure 2.** The full range of coccosphere geometry in *H. carteri*, *C. quadriperforatus* and *C. leptoporus*. (a)–(d) Histograms of coccosphere diameter (Ø) calculated for frequency bins of 1μm size. Note the different frequency scale in plot (d). (e)–(h) Number of coccoliths per cell ($C_N$) against Ø showing a strong and statistically significant (p<0.0001) positive relationship. The reduced major axis regression lines have slopes of *H. carteri* – 2.10 (bootstrapped 95 % confidence interval, CI [2.03, 2.17]; *C. quadriperforatus* – 2.11 (95 % CI [2.02, 2.20]); *C. leptoporus* – 3.01 (95 % CI [2.87, 3.14]); *C. braarudii* Gibbs et al., 2013 – 1.35 (95 % CI [1.27, 1.43]); *C. braarudii* Daniels et al., 2014 – 1.17 (95 % CI [1.09, 1.25]); *C. pelagicus* Daniels et al., 2014 – 1.50 (95 % CI [1.31, 1.67]). (i)–(l) Coccolith length ($C_L$) with Ø. (m)–(p) $C_L$ and Ø with data points coloured by $C_N$. For comparison purposes, we include data for *C. braarudii* and *C. pelagicus* that can be found in Gibbs et al. (2013) and Sheward et al. (2014) accompanying Daniels et al. (2014).

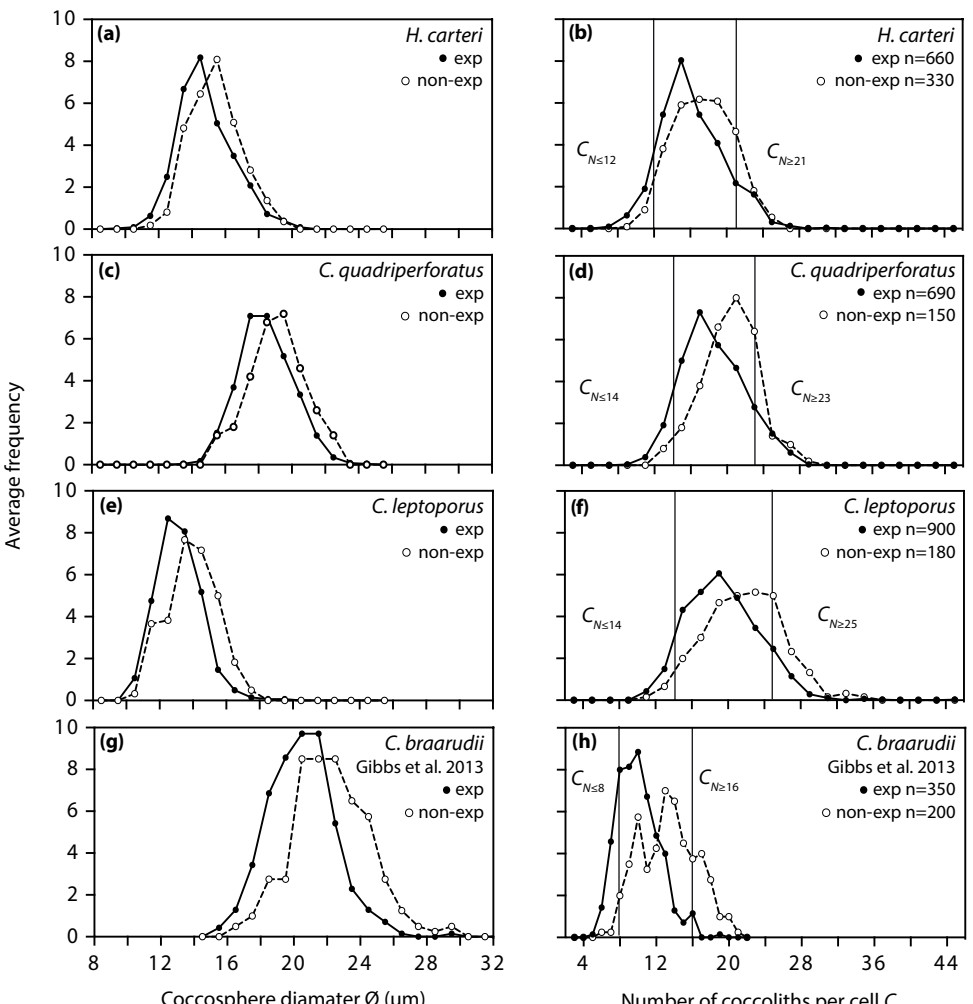

**Figure 3.** Frequency of coccosphere diameter (Ø) and number of coccoliths per cell ($C_N$) for experiment days in exponential growth (solid line) and experiment days no longer in exponential growth (dashed line), averaged across all temperature treatments. (a)–(f) *H. carteri*, *C. quadriperforatus*, and *C. leptoporus* data from this study. (g)–(h) is a reproduction of *C. braarudii* experiment data from Gibbs et al. (2013) SI Figure 1e. and 1.f for comparison purposes. The lines drawn on $C_N$ plots indicate cells that are recently divided and ready-to-

775 divide/non-dividing, based on the 10th and 90th percentiles of the complete species $C_N$ data shown in Fig. 2.

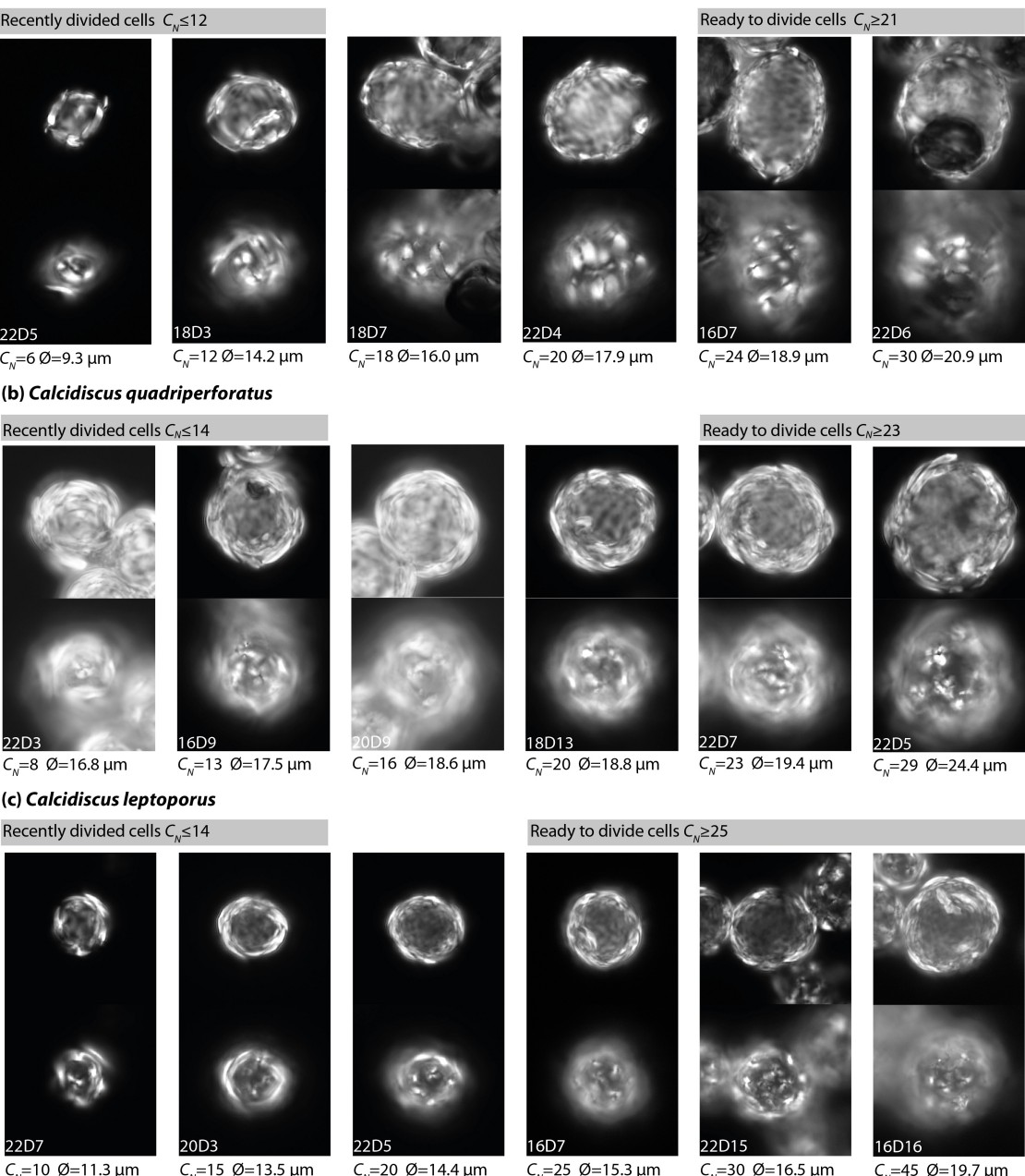

**(a) *Helicosphaera carteri***

Recently divided cells $C_N \leq 12$

Ready to divide cells $C_N \geq 21$

22D5 — $C_N$=6 Ø=9.3 µm
18D3 — $C_N$=12 Ø=14.2 µm
18D7 — $C_N$=18 Ø=16.0 µm
22D4 — $C_N$=20 Ø=17.9 µm
16D7 — $C_N$=24 Ø=18.9 µm
22D6 — $C_N$=30 Ø=20.9 µm

**(b) *Calcidiscus quadriperforatus***

Recently divided cells $C_N \leq 14$

Ready to divide cells $C_N \geq 23$

22D3 — $C_N$=8 Ø=16.8 µm
16D9 — $C_N$=13 Ø=17.5 µm
20D9 — $C_N$=16 Ø=18.6 µm
18D13 — $C_N$=20 Ø=18.8 µm
22D7 — $C_N$=23 Ø=19.4 µm
22D5 — $C_N$=29 Ø=24.4 µm

**(c) *Calcidiscus leptoporus***

Recently divided cells $C_N \leq 14$

Ready to divide cells $C_N \geq 25$

22D7 — $C_N$=10 Ø=11.3 µm
20D3 — $C_N$=15 Ø=13.5 µm
22D5 — $C_N$=20 Ø=14.4 µm
16D7 — $C_N$=25 Ø=15.3 µm
22D15 — $C_N$=30 Ø=16.5 µm
16D16 — $C_N$=45 Ø=19.7 µm

**Figure 4.** Light microscopy images illustrating the full range of cell geometry observed across all experiment days within cultures of (a) *H. carteri*, (b) *C. quadriperforatus*, and (c) *C. leptoporus* at 16-22 °C. The upper image of each pair shows the cross-sectional view of the cell from which coccosphere diameter and cell diameter are measured. The lower image of each pair shows a coccolith-focused view of the cell from which coccolith length is measured. Number of coccoliths per cell ($C_N$) and coccosphere diameter (Ø) are given for each cell. End-member geometries illustrating recently-divided and ready-to-divide cells are shown, based on their $C_N$ and Ø. Both exponential-phase and non-exponential-phase cultures will contain some recently-divided and some ready-to-divide cells, but the proportion (%) of each will differ depending on growth phase, as shown in Fig. 3 and Fig. 6. A reference code for the experiment day that the image was taken from is also given. 22D7 would be a cell from Day 7 of the 22 °C experiment as an example. All images are to the same scale.

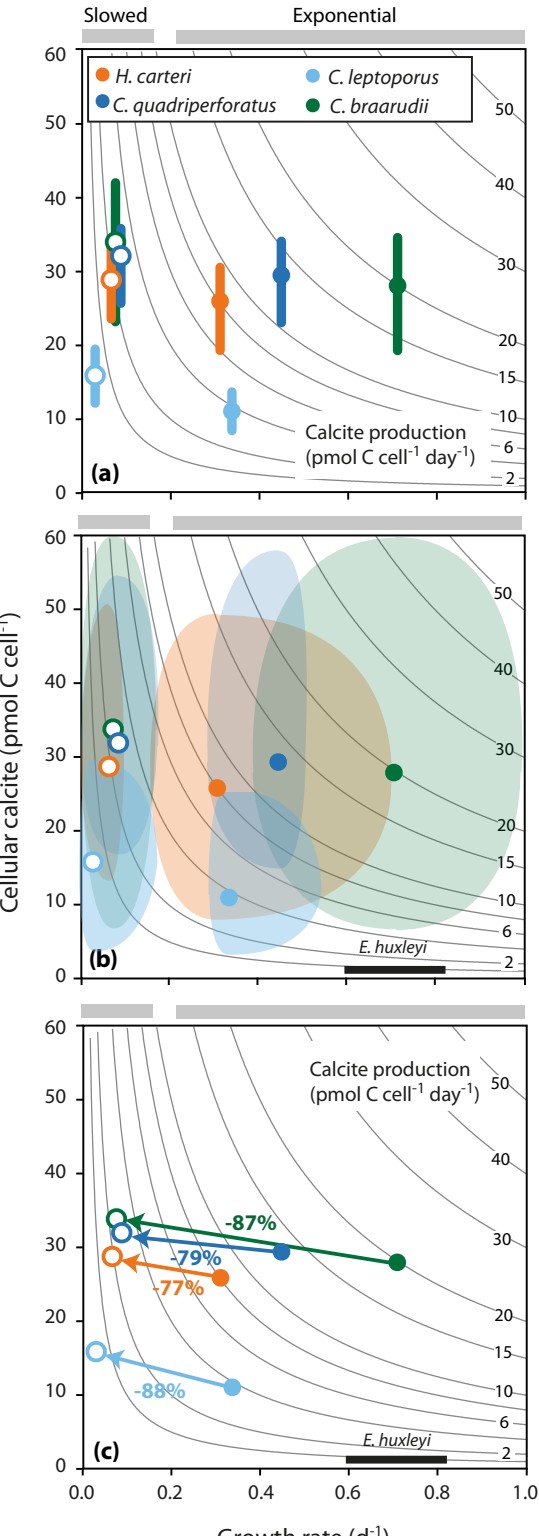

**Slowed**  **Exponential**

**(a)**

- ● H. carteri
- ● C. quadriperforatus
- ● C. leptoporus
- ● C. braarudii

Calcite production
(pmol C cell⁻¹ day⁻¹)

**(b)**

Cellular calcite (pmol C cell⁻¹)

E. huxleyi

**(c)**

Calcite production
(pmol C cell⁻¹ day⁻¹)

-87%
-79%
-77%
-88%

E. huxleyi

Growth rate (d⁻¹)

**Figure 5.** Calcification rates in *Coccolithus*, *Calcidiscus* and *Helicosphaera* at 22 °C. (a) mean and 25[th] to 75[th] percentile of cellular calcite for cultures dividing exponentially (mid-exponential-phase days, see Table S1; filled circles) and cultures no longer maintaining exponential growth (unfilled circles). (b) Range in cellular calcite, daily growth rates and calcite production observed across the experiment. (c) Percentage decrease in mean calcite production when cultures can no longer divide exponentially. The black box in (b) and (c) represents typical calcite production rates (~0.2-0.8 pmol C cell[-1] day[-1]) for *E. huxleyi* for comparison (Balch et al., 1996, Poulton et al., 2010).

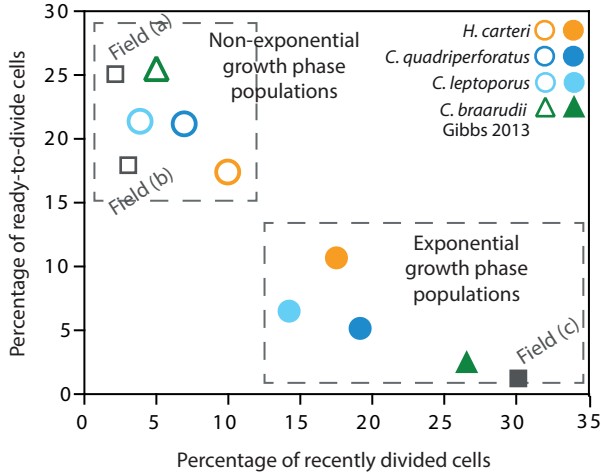

**Figure 6.** Contrasting exponential and non-exponential phase culture populations based on the percentage of recently divided and ready-to-divide cells within the population, as characterised by $C_N$ thresholds specific to each species (Fig. 3; Table 1). Mean percentages for exponential days are shown as filled data points and the mean non-exponential experiment day percentages are shown as unfilled data points. Also indicated (grey squares) are the characteristic percentages of three *Coccolithus* field population datasets presented in Gibbs et
al. (2013) - Field (a) is Scotland, Field (b) is Iceland non-bloom (both experiencing slowed growth), and Field (c) is Iceland bloom experiencing rapid growth.