# Peer review of "Physiology regulates the relationship between coccosphere geometry and growth-phase in coccolithophores"

_Biogeosciences, 2016_

## Referee Comment (RC1) · Anonymous Referee #1 · 29 Nov 2016

General comments: The manuscript is well written and overall clearly structured, although the results and discussion sections are not clearly separated with some results discussed in the results section and some new data ($\mu$, PIC) introduced in the discussion. The methods are provided in sufficient detail to allow reproduction, however, some additional information on how the data was treated in creating the figures could be useful, see specific comments. The results section needs to be carefully reread. There is a lack of consistency between described results and presented data (especially table 1), see specific comments. Furthermore, not all data described in the methods/presented in table 1 are discussed e.g. POC data. The manuscript addresses a relevant scientific question. The approach of using data from modern coccolithophores

to interpret the fossil record is highly relevant for understanding long-term trends and identifying possible future trends. In this light, intact fossil coccospheres pose an intriguing possibility to study the relationship between physiology and (palaeo-) environment and the authors add to this knowledge base by presenting data on coccosphere geometry of lesser studied (and genetically distant) species. Yet, I wonder what the range of applicability is. Do the authors have any knowledge on how common it is to find intact coccospheres rather than single coccoliths in the fossil record? Is there any data available for the species presented in the current manuscript? The conclusion that the trend observed by Gibbs et al. (2013) is a feature of coccolithophores as a whole is well founded. However, the hypothesis that coccosphere geometry can give information on population fitness is quite bold. In fact, the authors are very well aware of this fact and discuss the difference between growth phase and growth rate. Figure 5 nicely illustrates that an exponentially growing culture can be likened to a bloom situation in the field, whereas a non-bloom situation can look like a "stationary phase" culture. However, these situations are very short-lived in nature. The authors write themselves (L 301-303) "growth rates would not necessarily be expected to influence coccosphere geometry in the same way as a shift in growth phase". I would agree, but this limits the applicability of coccosphere geometry to the fossil record. To relate coccosphere geometry and nutrient availability in a geological setting, it is necessary to (at least) look at whether nutrient-induced changes in steady state i.e. continuous growth rates have an effect on average coccosphere geometry. The authors tried to address this by growing the species at different temperatures, which failed to induce a significant range in growth rate. The concept of "growth phase in the fossil record" is difficult because growth phases are of very short duration in nature. Over long time periods, steady state or maximum exponential growth rates become important. It is therefore misleading to relate growth phase to the fitness of a population (e.g. L 68; L 378) because all populations undergo different growth phases. The presented data is intriguing and propose a framework, which may prove useful in the future. However, the conclusion that this is a proxy for "population fitness" is a bit overstated (L 375-380)

and the discussion would benefit from including some thoughts on testing this hypothesis further e.g. by using chemostats. In a similar light, although the authors discuss this, in some parts of the text (see specific comments) it could be made clearer that it is only the average coccosphere size and CN that shifts, whereas the range is the same in both growth phases (Figure 3). Although Figure 4 looks very impressive, it creates the idea that in exponential phase "all cells" are small, whereas in stationary phase "all cells" are large. However, figure 3 makes it clear that both stages span the same size range and there is only a slight (though significant) shift in the average size.

Specific comments: L 14-16: "however, to realize the potential of this archive requires an understanding" needs to be rephrased. L 36: why do the authors use a threshold size of 63 $\mu$m, instead of e.g. 200 $\mu$m, the threshold for microplankton? Does this relate to the max. size of coccolithophores? L 37-38: what do the authors mean by "the biomass that sustains the wider diversity of marine life at higher trophic level"? I would argue that the (genetic) diversity of photosynthesizing plankton surpasses that of higher trophic levels. L 93: should it be "classified into separate families" rather than "in"? L 105: please check the calculation of daily photon flux; it should be half that value; maybe the calculation was carried out using 24-h light? L 106: the authors could consider mentioning up front that the different temperatures failed to give a significant range in cell division rates and therefore all data was pooled. L 108-110: I would add a table as supplementary information with $\mu$ (rather than just mentioning the range in the discussion) and coccosphere geometry parameters for each temperature experiment. The authors state that there were no (significant?) differences among the temperature treatments. However, they have used only two replicates for each temperature. How was this tested? L 114-115: please give nitrate and phosphate concentrations of the medium as this is important for interpreting final cell concentrations. The authors give references to the K/20-medium, but following Daniels et al. (2014), nitrate concentrations would be 28.8 $\mu$M and phosphate 1.8 $\mu$M, following Gerecht et al. (2014), concentrations would only be 16 $\mu$M nitrate and 1 $\mu$M phosphate. Šupraha et al. (2015) presented data on the same strain of Helicosphaera used in the present experiment

which entered stationary phase at ca. half the cell concentration (15500 cells mL-1) as in the current study (growing on 1 $\mu$M initial phosphate; I therefore assume phosphate concentration was 1.8 $\mu$M? L 115-119: Daniels et al. (2014) grew dilute batch cultures and harvested in exponential phase, whereas the authors have used the same medium (?), but harvested in stationary phase. In fact, the cultures reached much higher cell concentrations (for C. braarudii 25.000 cells mL-1 vs. max. 8.700 in Daniels et al. (2014)). The authors do not give information on the carbonate chemistry, but based on the high final cell concentrations, I would assume that there was a significant consumption of DIC (for comparison, in Gerecht et al. (2014), C. braarudii reached max. cell concentrations of 17550 cells mL-1, which reduced DIC down to 1200 $\mu$M. A rough calculation and assuming a proportional response, at 25000 cells mL-1, DIC would be reduced to ca. 900 $\mu$M which is assumedly limiting for growth (Bach et al., 2013). How much can this really be compensated for by passive diffusion into a bottle and then into the medium, which are both presumably slow processes? Similarly, Šupraha et al. (2015) presented data on a Mediterranean strain of Helicosphaera that entered stationary phase at 41.000 cells mL-1 vs. the 30.000 in this study. At 41.000 cells mL-1, carbonate chemistry was severely altered (and PIC quota of the single coccoliths was affected). A significant change in carbonate chemistry and ensuing DIC limitation does not directly affect the conclusions of the manuscript, also because PIC quota was not directly measured, but inferred from coccolith length measurements. However, L 115-119 is misleading in suggesting that carbonate chemistry was not affected as no data is presented to confirm this. On the contrary, the available data indicate that there was a significant impact (as outlined above). L 278: POC production could have been affected by DIC limitation. L 127: assumedly, formaldehyde was also added to Helicosphaera cultures for size measurements? Although the coccosphere is probably relatively stable, formaldehyde may lead to shrinking of the cell membrane, which could influence morphology measurements. Did the authors consider/check this? L 135: no need to mention that half of the filters were stored for SEM, if the data is not presented. Did the authors do any control measurements under SEM? L 143: If coccosphere size

is used as a (realistic) proxy for cell size, the terminology should be used more clearly. In the methods, coccosphere and cell size are presented as two different parameters, whereas in the abstract "cell size" is used more loosely. L 170: as daily growth rates ranged between max. and zero, it is confusing to write "resulted in a modest range of daily and mean exponential growth rates"; remove daily. L 170: which days were used to calculate mean exponential growth rates? L 182: none of the values reported in the text for coccosphere diameter correspond to those presented in table 1 i.e. max value for H. carteri is 21, not 15 $\mu$m. L 187: mean CN for C. braarudii is 14, not 11-12 $\mu$m. L 188-189: move information on large coccosphere in C. leptoporus up one sentence. L 211: according to table 1, CL varies by 8.0 $\mu$m in C. quadriperforatus. L 217-218: I agree that there is no relationship between CN and cell diameter. However, I do not understand this sentence. The range in both coccosphere diameter and CL is very broad. Also, the cultures are not synchronized in regard to cell division as large cells ("about to divide") are also present in exponential phase and small cells ("just divided") also in stationary phase; only the peak of the mean shifts (see figure 3). L 235-239: the way this is phrased is misleading; it sounds as if "all cells" in stationary phase contain numerous coccoliths whereas you still have small, recently-divided cells. L 254: in the results the shift in coccosphere diameter is 0.55-0.7 $\mu$m (1.75 $\mu$m for C. braarudii (L 229); this does not amount to ca. 2 $\mu$m. According to my calculations, using mean diameter presented in L 227-228 and the above mentioned increases, there is a 3% increase in Helicosphaera coccosphere diameter, 4-5% in Calcidiscus and 9% in Coccolithus. Here, however, the authors write 10-12%. If the authors calculated differently/used different values, this needs to be made clearer in the text. L 300: in results, max. $\mu$ for H. carteri is listed as 0.45 d-1. L 300-303. This argument is not convincing. If coccosphere geometry is to be a tool for looking at long-term (as this is the fossil record) changes in growth, then it is necessary to look at reduced physiological rates due to e.g. lower nutrient availability. It is not possible to look at long-term shifts in growth phase in the fossil record. If there is an increased percentage in large cells in a certain sediment layer, what does this mean? That this was a time of "nutrient

limitation" so that more cells grew larger and stopped dividing? Although this hypothesis is intriguing, it is necessary to look for a link between reduced physiological rates and larger cells if this is to be applied as a proxy to the fossil record. L 310: in this study, coccolith length is independent of growth phase so this factor of coccosphere geometry should be removed here. L 416: Gerecht et al. (2015) show that growth rate determines calcite production in C. braarudii: PIC production is reduced by ca. 50% due to the 50% reduction in growth rate. Table 1: is this all combined data i.e. exponential and stationary phase? Table 1: check values Max PIC for C. braarudii (lower than Mean) Table 1: there is no mention in the methods as to how POC per cell was calculated, nor is it presented in the results or discussed in the rest of the paper. Figure 2: check frequency values for right y-axis. Figure 3: How is "early stationary-phase growth" defined i.e. which daily growth rate is still considered exponential? Figure 4 provides convincing evidence that coccosphere geometry data can be obtained by POL. I wonder, however, why SEM was not used? This would supply more accurate measurements of coccosphere diameter and especially of CL and equally adequate estimates of CN. It would not provide cell diameter, but the authors do not discuss cell diameter in the text. Cell diameter should be mentioned in the results as it is included in table 1, even if only as a short sentence e.g. cell diameter followed the same trend as coccosphere diameter (?). The authors present calcite production rates in Figure 6. I would appreciate a sentence on how this was calculated in the Methods section i.e. was this an average of all PIC contents x growth rates in exponential phase or just from one specific day? Likewise for the "stationary phase"-values.

Technical corrections: Please check the bibliography closely. I have found at least 3 errors: de Vargas 2004 and Keller et al. 1987 are missing, whereas Young 2003 is in the citation list, but not cited in the text. Ziveri 2007: check journal name L 420: (Toweius) pertusus needs to be cursive

Bach, L.T., Mackinder, L.C.M., Schulz, K.G., Wheeler, G., Schroeder, D.C., Brownlee, C., Riebesell, U., 2013. Dissecting the impact of CO2 and pH on the mechanisms of

photosynthesis and calcification in the coccolithophore Emiliania huxleyi. New Phytologist 199, 121-134. Daniels, C.J., Sheward, R.M., Poulton, A.J., 2014. Biogeochemical implications of comparative growth rates of Emiliania huxleyi and Coccolithus species. Biogeosciences 11, 6915-6925. Gerecht, A.C., Šupraha, L., Edvardsen, B., Probert, I., Henderiks, J., 2014. High temperature decreases the PIC / POC ratio and increases phosphorus requirements in Coccolithus pelagicus (Haptophyta). Biogeosciences 11, 3531-3545. Gerecht, A.C., Šupraha, L., Edvardsen, B., Langer, G., Henderiks, J., 2015. Phosphorus availability modifies carbon production in Coccolithus pelagicus (Haptophyta). J. Exp. Mar. Biol. Ecol. 472, 24-31. Gibbs, S.J., Poulton, A.J., Bown, P.R., Daniels, C.J., Hopkins, J., Young, J.R., Jones, H.L., Thiemann, G.J., O'Dea, S.A., Newsam, C., 2013. Species-specific growth response of coccolithophores to Palaeocene-Eocene environmental change. Nature Geoscience 6, 218-222. Šupraha, L., Gerecht, A.C., Probert, I., Henderiks, J., 2015. Eco-physiological adaptation shapes the response of calcifying algae to nutrient limitation. Scientific Reports 5(16499).

---

## Referee Comment (RC2) · Anonymous Referee #2 · 20 Dec 2016

The manuscript 'Physiology regulates the relationship between coccosphere geometry and growth-phase in coccolithophores' authored by Sheward and co-workers presents novel results, is very well written, has a well reasoned rationale and an overall well structured text, and clear figures. The study addresses an interesting topic, namely the relationship between coccosphere architecture and coccolithopores' (exponential versus stationary) growth phases in four species of extant coccolithophores. The data presented are new and of good quality and do support the conclusions drawn by the authors. In addition, the manuscript is presented in a way that will be also accessible to non-specialists, which is an added value for publication in a multidisciplinary journal such as Biogeosciences. I therefore recommend this manuscript for publication in

Biogeosciences provided that the specific points listed below are addressed.

Specific points:

(1) Much emphasis is put on the relevance of this study for the investigation/interpretation of coccolithophore geometries in the fossil record as a proxy for coccolithophore growth phase. However, in the introduction the authors state that intact fossil coccospheres can be found in '. . .exceptionally well-preserved sedimentary deposits. . .'. I would therefore suspect that the application of this growth phase proxy is perhaps useful only in a very limited number of settings and of a few geologic periods where/when intact fossil coccospheres are found. That being the case, the statements about the relevance of this study for the interpretation of the palaeorecords (also in the conclusions) should be toned down, at least in the terms used by the authors. Given that this is not the main reason why this is a valuable piece of work, these statements could be toned down without affecting the relevance and novelty of the study. (2) Often times in the manuscript it is stated that results are statistically significant, but a section in the methods that specifically presents the statistical approaches used in this study is missing and should be added. Also, data analysis could benefit from some (bootstrap?) outlier analysis, specifically when different properties of the coccolithophore geometry are regressed against one another (e.g., Fig. 2e-h). This would certainly improve the analysis of the high quality (and rich) dataset presented in this study.

---

## Author Comment (AC1) · 24 Jan 2017

We kindly thank both reviewers for their positive, insightful and constructive comments on our manuscript and we are pleased that the manuscript was considered to be well-written, clearly structured and accessible to non-specialists, with well-reasoned rationale and a high-quality dataset. We have carefully considered all comments and suggestions and hope that we have satisfactorily addressed each point raised. The main points that we have addressed are as follows (see details in specific responses):

Applicability and significance of work: Both Reviewers questioned how common the occurrence of coccospheres is in the fossil record and therefore how broadly applicable our results may be. Our wider research demonstrates that coccospheres are uncommon but not so rare (documented from a range of ages, ocean basins and latitudes) that our approach would only be applicable in very few cases. We have now emphasised our comments on applicability in the introduction, discussion and conclusions.

Correction of Table 1: We thank Reviewer #1 for their diligence in identifying inconsistencies between the data in Table 1 and in the text. We have checked all reported values for accuracy and have changed values in Table 1 and in the text where necessary.

Additional Supplementary Table: In light of suggestions from Reviewer #1, we have prepared a Supplementary Table to report growth rates and minimum, mean and maximum coccosphere geometry for Exponential-phase and Non-exponential-phase days separately to complement the 'full dataset' coccosphere geometry summarised in Table 1.

Growth phase, growth rate and fitness comments: We have carefully considered the comments of Reviewer #1 concerning interpreting growth information in the fossil record based on growth phase results rather than growth rate results. In light of specific comments, we have adjusted our phrasing for clarity and to avoid misleading word choices in several places (Ln 67-68; Ln 368; Ln 375-379). We have also added additional comments in the discussion on the value of other experimental approaches in addressing growth rate and coccosphere geometry in future investigations to further advance our work.

Statistical methods: We have added a 'Statistical Methods' sub-section to the Methods following the suggestion of Reviewer #2, including bootstrap analysis of our linear regression analysis, which we agree improves the high quality of our extensive dataset.

Below are the detailed author responses to the individual comments made by Reviewer #1 and #2.

Response to Reviewer #1

General comments:

1: The manuscript is well written and overall clearly structured, although the results and discussion sections are not clearly separated with some results discussed in the results section and some new data ($\mu$, PIC) introduced in the discussion.

Author response: We introduce 'new' analysis into our discussion about the wider implication of our results for calcification rates, building on previous discussion about coccosphere geometry and growth phase. The exponential growth rate data is presented in the Results: Sect. 3.1 – Growth rates. We now additionally include the range of growth rates during exponential and non-exponential growth days in this section and summarise this is a new Supplementary Table. The PIC per cell and PIC production data is presented for the first time in Sect. 4.4 as it is not addressing the primary aim of the study. However, it will now be reported within the results section.

2: There is a lack of consistency between described results and presented data (especially table 1)

Author response: We apologise for the errors in Table 1 and have now checked for and corrected any misreported values in the text and Table 1, see specific comments and revised data in Table 1.

3: Not all data described in the methods/presented in table 1 are discussed e.g. POC data.

Author response: We have removed POC data from Table 1.

4: I wonder what the range of applicability is. Do the authors have any knowledge on how common it is to find intact coccospheres rather than single coccoliths in the fossil record? Is there any data available for the species presented in the current manuscript?

Author response: The presence of coccospheres in sediments is considered atypical, though the authors of this study are involved in a project specifically investigating Paleogene coccospheres (manuscript in preparation) that targets sediments with higher

numbers of coccospheres than usual, ∼20 to >300 per standard smear slide (see Gibbs et al., 2013 and Bown et al., 2014). So far, we have documented and measured 4,388 individual coccospheres spanning more than 40 Paleogene species and 9 families, including important extant families. Therefore, coccosphere geometry is potentially relatively widely applicable in the geological record. We explain our reasoning for choosing Calcidiscus and Helicosphaera species to supplement Coccolithus data in the Introduction (L 76-95). Coccolithus is an exceptionally long-lived genus, first originating ∼66 Ma, and the species Coccolithus pelagicus has arguably existed for roughly the same length of time. It represents a large proportion of our fossil coccosphere dataset (∼1800 coccospheres to date; Sheward et al., in prep). In the Paleogene, where we are currently focusing our fossil coccosphere studies, Calcidiscus and Helicosphaera are less abundant contributors to the overall assemblage and we do not have detailed fossil coccosphere data for them, although we do have some Miocene fossil coccospheres of Helicosphaera and Calcidiscus spp. coccospheres from the Middle Eocene. However, these genera become increasingly abundant into the Neogene and towards the present. Their robust coccolith morphologies and coccosphere architectures give them a comparably high preservation potential, similarly to Coccolithus. The majority of the fossil coccospheres we see are extinct species, but information about taxa in the same genus or family is still highly valuable. In terms of applicability outside these genera, we detail in Sect. 4.3, L 346-354 how our methodology using CN thresholds to investigate qualitative variability in growth phase could be applied to any extinct taxa provided there was an abundant fossil coccosphere geometry dataset to use.

5: The hypothesis that coccosphere geometry can give information on population fitness is quite bold. In fact, the authors are very well aware of this fact and discuss the difference between growth phase and growth rate. The authors write themselves (L 301-303) "growth rates would not necessarily be expected to influence coccosphere geometry in the same way as a shift in growth phase". I would agree, but this limits the applicability of coccosphere geometry to the fossil record. To relate coccosphere geometry and nutrient availability in a geological setting, it is necessary to (at least)

[Figure]

look at whether nutrient-induced changes in steady state i.e. continuous growth rates have an effect on average coccosphere geometry. The authors tried to address this by growing the species at different temperatures, which failed to induce a significant range in growth rate. The concept of "growth phase in the fossil record" is difficult because growth phases are of very short duration in nature. Over long time periods, steady state or maximum exponential growth rates become important. It is therefore misleading to relate growth phase to the fitness of a population (e.g. L 68; L 378) because all populations undergo different growth phases. The discussion would benefit from including some thoughts on testing this hypothesis further e.g. by using chemostats.

Author response: We did not intend for our phrasing to lead readers to consider our discussion as presenting a 'proxy for fitness'. Biological fitness describes population reproductive success that is usually measured as growth rate in experimental phytoplankton studies (e.g., Collins et al., 2013) and cell size and calcification are also considered to be important fitness-related traits (e.g., Lohbeck et al., 2012; Benner et al., 2013; Jin et al., 2013). As our results produced no obvious correlation between growth rate and coccosphere geometry, we currently cannot use coccosphere geometry to estimate a specific growth rate in field populations or fossil assemblages. However, as we say in L 59-61, growth phase describes two different states, one of which is of slowed to zero growth caused by nutrient paucity. Therefore, analysing the coccosphere geometry of populations/assemblages and its variability through time using the method we present identifies intervals when population coccosphere geometry is more representative of such a slowed growth state or conversely rapid, exponential growth. This type of growth information is not measurable from typical coccolithophore remains in the fossil record (usually just loose coccoliths), so it is the first indicator of growth of any form from intact coccospheres. In light of the reviewers comments, we have rephrased L67-68 and L375-379 so that they do not imply growth phase as a fitness trait, rather that interpretations of growth phase in field or fossil population is an important first step in retrieving any sort of indication of fitness traits in the fossil coccolithophore record. We now add a section in the discussion following on from the end of L368 to

address the valuable additional perspective that using continuous or semi-continuous approaches would provide, and that this approach should be used to specifically investigate the effect of a range of steady-state growth rates on coccosphere geometry, which would greatly strengthen future investigations of growth in the fossil record.

6: It could be made clearer that it is only the average coccosphere size and CN that shifts, whereas the range is the same in both growth phases (Figure 3). Although Figure 4 looks very impressive, it creates the idea that in exponential phase "all cells" are small, whereas in stationary phase "all cells" are large. However, figure 3 makes it clear that both stages span the same size range and there is only a slight (though significant) shift in the average size.

Author response: Fig. 4 is largely to demonstrate visually that coccosphere geometry is very variable within a species, as typically only mean values and standard deviations are referred to in published studies and we have rephrases the caption of Fig. 4 to make this very clear. By referring to the 'recently-divided' and 'ready to divide' types of cells, we are highlighting to the reader the 'end-member' type of coccosphere geometries used in estimating the proportion of the population that has recently undergone cell division, which will be higher during exponential phase when the time between cell divisions is shorter. We show this in Fig. 5 for exponential days and non-exponential days of growth in these culture experiments and selected field samples experiencing characteristics of 'bloom' and 'non-bloom' conditions. To improve clarity, L252-254 have been re-worded to more clearly state that there is an increased proportion of cells in larger Ø or CN size classes respectively, and a shift in mean Ø and CN (which has been tested statistically). Section 3.3 of the results describing Fig. 3 is now more explicit, making it much clearer when changes in mean Ø or CN are referred to, relative to overall changes in the histogram shapes. We have also added that minimum Ø and CN shifts towards larger values in the non-exponential-phase data (Ø increase of 0.77 to 2.29 $\mu$m, representing a species-specific 6.2 to 24.5 % increase on exponential-phase minimum Ø, and a CN increase of 1 to 4 on minimum exponential CN, representing a

species-specific increase of 10 to 50% on minimum exponential CN). This data is now shown in the Supplementary Table.

Specific comments:

7: L 14-16 - "however, to realize the potential of this archive requires an understanding" needs to be rephrased.

Author response: Re-phrased as "However, to realize the full potential of this archive for (paleo-)biology and biogeochemistry requires an understanding..."

8: L 36 - why do the authors use a threshold size of 63 $\mu$m, instead of e.g. 200 $\mu$m, the threshold for microplankton? Does this relate to the max. size of coccolithophores?

Author response: Nannoplankton and nannofossils are defined by Lohmann (1909) and Young et al. (1997) as ranging between 2 and 63 $\mu$m and the 63 $\mu$m threshold is employed by micropaleontologists wishing to physically separate the calcareous nannoplankton fraction (coccolithophores and other nannoplankton) from other microfossil groups within a sediment sample. The nannoplankton range is considered by some authors to be between 2 and 20 $\mu$m (e.g., Sieburth et al., 1978; Finkel et al., 2010). However, coccolithophore cell/coccosphere sizes regularly exceed this, with the approximate maximum coccosphere size in modern coccolithophores ∼53 $\mu$m in Coronosphaera sp. (O'Brien et al., 2013). Using a 200 $\mu$m threshold would be suitable for phytoplankton in general, but may be misleading about the maximum size of coccolithophores.

9: L 37-38 - what do the authors mean by "the biomass that sustains the wider diversity of marine life at higher trophic level"? I would argue that the (genetic) diversity of photosynthesizing plankton surpasses that of higher trophic levels.

Author response: We have revised the phrasing for clarity. We simply meant that the biomass produced by photosynthesizing plankton fuels all the marine life at higher trophic levels.

10: L 93 - should it be "classified into separate families" rather than "in"?

Author response: We have changed the sentence to now read "classified into separate families"

11: L 105 - please check the calculation of daily photon flux; it should be half that value; maybe the calculation was carried out using 24-h light?

Author response: The reviewer is correct that daily photon flux should be half of the value stated, it was calculated for a 24 hour 'day' without accounting for the 'daylight' period being only 12 h of that. Values of 75 – 90 $\mu$mol photons m-2 s-1, under a 12 h photoperiod, 75 = 3.24 and 90 = 3.89 mol photons m-2 d-1.

12: L 106 - the authors could consider mentioning up front that the different temperatures failed to give a significant range in cell division rates and therefore all data was pooled. 13: L 108-110 - I would add a table as supplementary information with $\mu$ (rather than just mentioning the range in the discussion) and coccosphere geometry parameters for each temperature experiment.

Author response: All of the $\mu$ and coccosphere geometry parameters for every experiment day are provided as a pangeae.de datafile. For completeness, a new Supplementary Table now presents mean, min and max CL, CN, Ø, Θ for exponential and non-exponential days and mean exponential $\mu$ for each temperature experiment.

Cont. from above: The authors state that there were no (significant?) differences among the temperature treatments. However, they have used only two replicates for each temperature. How was this tested?

Author response: We state L 299-300 that "exponential growth rates were not that sensitive to the temperature range we applied (C. quadriperforatus $\mu$exp=0.30-0.44 d-1; C. leptoporus $\mu$exp=0.31-0.44 d-1; H. carteri $\mu$exp=0.28-0.45 d-1)." The reported values are the minimum and maximum mean exponential growth rates across the four temperature conditions for each species. We do not statistically compare the two replicates at each temperature. When coccosphere geometry data from exponential experiment days was analysed against the mean exponential growth rate of the experiments there was no statistical correlation that would suggest an obvious relationship between growth rate and any coccosphere geometry parameter that was measured.

14: L 114-115 - please give nitrate and phosphate concentrations of the medium as this is important for interpreting final cell concentrations. The authors give references to the K/20-medium, but following Daniels et al. (2014), nitrate concentrations would be 28.8 $\mu$M and phosphate 1.8 $\mu$M, following Gerecht et al. (2014), concentrations would only be 16 $\mu$M nitrate and 1 $\mu$M phosphate. Šupraha et al. (2015) presented data on the same strain of Helicosphaera used in the present experiment which entered stationary phase at ca. half the cell concentration (15500 cells mL-1) as in the current study (growing on 1 $\mu$M initial phosphate; I therefore assume phosphate concentration was 1.8 $\mu$M?

Author response: We have modified L113-115 to now give details of the medium nutrients, as follows, and removed the reference to Gerecht et al. (2014) to avoid confusion as they used different concentrations: "...added to 350 ml of sterilised and filtered natural seawater enriched with 28.8 $\mu$M nitrate and 1.8 $\mu$M phosphate (lower-nutrient K/20 medium, modified from Keller et al., 1987, following Langer et al., 2006 and Daniels et al., 2014).

15: L 115-119 - Daniels et al. (2014) grew dilute batch cultures and harvested in exponential phase, whereas the authors have used the same medium (?), but harvested in stationary phase. In fact, the cultures reached much higher cell concentrations (for C. braarudii 25.000 cells mL-1 vs. max. 8.700 in Daniels et al. (2014)).

Author response: Yes, we used the same growth medium and the difference in maximum cell concentrations arises because Daniels et al. (2014) sampled in mid-exponential phase growth (to ensure truly exponential phase growth) not at maximum cell density (where cells are no longer in exponential phase growth under a batch culture approach). Samples for cell counts and LM coccosphere geometry measurements in Coccolithus (Gibbs et al., 2013), Calcidiscus and Helicosphaera were harvested daily throughout the experiment, not just in the stationary phase.

16: The authors do not give information on the carbonate chemistry, but based on the high final cell concentrations, I would assume that there was a significant consumption of DIC (for comparison, in Gerecht et al. (2014), C. braarudii reached max. cell concentrations of 17550 cells mL-1, which reduced DIC down to 1200 $\mu$M. A rough calculation and assuming a proportional response, at 25000 cells mL-1, DIC would be reduced to ca. 900 $\mu$M which is assumedly limiting for growth (Bach et al., 2013). How much can this really be compensated for by passive diffusion into a bottle and then into the medium, which are both presumably slow processes? Similarly, Šupraha et al. (2015) presented data on a Mediterranean strain of Helicosphaera that entered stationary phase at 41.000 cells mL-1 vs. the 30.000 in this study. At 41.000 cells mL-1, carbonate chemistry was severely altered (and PIC quota of the single coccoliths was affected). A significant change in carbonate chemistry and ensuing DIC limitation does not directly affect the conclusions of the manuscript, also because PIC quota was not directly measured, but inferred from coccolith length measurements. However, L 115-119 is misleading in suggesting that carbonate chemistry was not affected as no data is presented to confirm this. On the contrary, the available data indicate that there was a significant impact (as outlined above).

Author response: We have now rephrased L 115-119 to avoid implying that the carbonate chemistry was unaltered, although we cannot report on specific changes as the carbonate system was not monitored. We thank the reviewer for pointing out that DIC could have changed over the course of the experiments and influenced growth and/or calcification and we agree that any change in the carbonate chemistry would not affect the conclusions that we draw from our results. The culturing flasks that we used have membrane caps that allow faster diffusive gas exchange than would occur in fully-sealed flasks as well as aerated and mixed each flask daily. Our light microscopy

work did not show any evidence of coccolith changes e.g., malformation, that might suggest a detrimental affect of carbonate chemistry parameters on coccolith calcification during our experiments (see LM images in Figure 4).

17: L 127 - assumedly, formaldehyde was also added to Helicosphaera cultures for size measurements? Although the coccosphere is probably relatively stable, formaldehyde may lead to shrinking of the cell membrane, which could influence morphology measurements. Did the authors consider/check this?

Author response: Formaldehyde was only added to samples for cell counts and not to the samples for morphological analysis, which is why the formaldehyde is not mentioned in the Methods section 2.3: Coccosphere geometry.

18: L 135 - no need to mention that half of the filters were stored for SEM, if the data is not presented. Did the authors do any control measurements under SEM?

Author response: No SEM measurements were done for control and the comment is removed from the text.

19: L 143 - If coccosphere size is used as a (realistic) proxy for cell size, the terminology should be used more clearly. In the methods, coccosphere and cell size are presented as two different parameters, whereas in the abstract "cell size" is used more loosely.

Author response: The measurement of the interior of the coccosphere is termed as 'cell diameter' and the measurement of the exterior of the coccosphere is termed 'coccosphere diameter' (this is the measurement that would be taken from an SEM image), as shown on Figure 1 and described in the Figure caption, with the assumption that the cell fully fills the coccosphere. The method text L143 now says "...and cell size ($\Theta$; size excluding calcite covering that is assume to be equivalent to cell diameter)..." and the use of cell size in the abstract has been replaced with coccosphere size.

20: L 170 - as daily growth rates ranged between max. and zero, it is confusing to write "resulted in a modest range of daily and mean exponential growth rates"; remove daily

Author response: By this sentence we mean that both daily growth rates during the exponential days of the experiment (growth rates calculated between consecutive days) and the mean exponential growth rate calculated across all exponential days of growth show a modest range. The sentence has been changed to improve clarity.

21: L 170 - which days were used to calculate mean exponential growth rates?

Author response: Days during which cultures were growing at an exponential rate are as follows: C. quadriperforatus = up to and including Day 11 (22 °C and 20 °C), Day 16 (18 °C), Day 17 (16 °C); C. leptoporus = up to and including Day 15 (20 °C), Day 11 (20 °C), Day 17 (18 °C), Day 18 (16 °C); H. carteri = up to and including Day 11 (22 °C), Day 7 (20 °C), Day 15 (18 °C), Day 13 (16 °C), C. braarudii = up to and including Day 11. Mean exponential growth rates ($\mu$) for each temperature experiment were calculated from cell densities where $\mu$ = (ln N1 –ln N0)/d, and N1 and N0 are cell concentrations at the beginning and end of the exponential phase and d is the duration of the exponential phase in days. A statement to this effect has been added to the end of Section 2.2, L130.

22: L 182 - none of the values reported in the text for coccosphere diameter correspond to those presented in table 1 i.e. max value for H. carteri is 21, not 15 $\mu$m.

Author response: We apologise for the minor errors in Table 1. All of the data in Table 1 has now been checked for accuracy and the values in Table 1 and the text corrected where appropriate. L182 now reads: "A considerable range in Ø is seen in all species; 13.8 $\mu$m to 24.4 $\mu$m in C. quadriperforatus, 9.4 $\mu$m to 20.9 $\mu$m in H. carteri, and 10.0 $\mu$m to 19.7 $\mu$m in C. leptoporus. This is a comparable Ø range to C. pelagicus (11.7 to 20.8 $\mu$m) but slightly less than the Ø range observed in C. braarudii (13.7 to 29.7 $\mu$m)."

23: L 187 - mean CN for C. braarudii is 14, not 11- 12 $\mu$m.

Author response: L187 now reads "In contrast, Coccolithus cells more typically have 11-14 coccoliths per cell, up to a maximum of ∼20 coccoliths." Coccolithus includes

both C. pelagicus and C. braarudii.

24: L 188-189 - move information on large coccosphere in C. leptoporus up one sentence.

Author response: Sentence has been moved.

25: L 211 - according to table 1, CL varies by 8.0 $\mu$m in C. quadriperforatus.

Author response: The minimum CL reported for C. quadriperforatus in Table 1 was incorrect and has now been changed in the Table (min CL = 5.67). A range in CL of 6.0 $\mu$m as stated in L211 is therefore correct.

26: L217-218 - I agree that there is no relationship between CN and cell diameter. However, I do not understand this sentence. The range in both coccosphere diameter and CL is very broad. Also, the cultures are not synchronized in regard to cell division as large cells ("about to divide") are also present in exponential phase and small cells ("just divided") also in stationary phase; only the peak of the mean shifts (see figure 3).

Author response: We believe that the reviewer here meant to say "I agree that there is no relationship between CL and cell diameter" rather than CN, which does of course show a relationship. The statement 'relatively restricted ranges of Ø and CL" refers to the larger range in CL and Ø that results when you combine data from multiple populations, which does produce a CL Ø relationship, as shown in Gibbs et al. (2013). We have removed the remark "...cell division is fully synchronized across cells...". This referred to sampling at the same time every day to minimize variability in coccosphere geometry resulting from the cell division cycle, such that the population should be made up of a broadly similar proportion of cells in different stages of the cell division cycle between each (exponential-phase) day. L216 now reads "In our clonal populations, cells have a relatively restricted range in Ø and CL that have no statistically-significant relationship (Fig. 2i-l)."

27: L 235-239 - the way this is phrased is misleading; it sounds as if "all cells" in

stationary phase contain numerous coccoliths whereas you still have small, recently-divided cells.

Author response: L235-239 have been re-phrased to say explicitly that we report changes in mean Ø and CN and that overall the distribution of cells within the population shifts towards larger size classes, as shown by the histograms in Figure 3 (no change in maximum Ø or CN).

28: L 254 - in the results the shift in coccosphere diameter is 0.55-0.7 $\mu$m (1.75 $\mu$m for C. braarudii (L 229); this does not amount to ca. 2 $\mu$m. According to my calculations, using mean diameter presented in L 227-228 and the above mentioned increases, there is a 3% increase in Helicosphaera coccosphere diameter, 4-5% in Calcidiscus and 9% in Coccolithus. Here, however, the authors write 10-12%. If the authors calculated differently/used different values, this needs to be made clearer in the text.

Author response: Here, L254 should have read "...up to $\sim$2 $\mu$m" but we agree that this would present an over-approximation so L252-254 now reads: "Across all four species investigated, the transition from exponential into non-exponential phase growth was clearly associated with a shift towards an increased abundance of cells with greater CN (mean CN increased by 1-3) and larger coccosphere sizes (mean Ø increased by 0.6 $\mu$m in H. carteri and C. quadriperforatus, 0.9 $\mu$m in C. leptoporus and 1.3 $\mu$m in C. braarudii). This represents a significant increase of 4 to 7% on exponential-phase mean Ø and a significant increase of 10 to 27 % on exponential phase mean CN (t-test, p<0.0001)."

29: L 278 - POC production could have been affected by DIC limitation.

Author response: We say that POC per cell (not POC production) could increase under nutrient limited conditions but we assume that this is what the reviewer is referring to. In this case, referring to the data of Šupraha et al. (2015) for two H. carteri strains, the POC cell-1 of their Atlantic strain increased 43.6% under P-limitation (no significant change in DIC, 1220 to 1159 $\mu$mol kg-1) and the POC of their Mediterranean strain that

did experience substantial DIC change (1292 to 639 $\mu$mol kg-1) showed a POC cell-1 increase of 46.7 %, which is not vastly different (their Supplementary Table 2 and Table 3). Moolna and Rickaby (2012) also report Gephyrocapsa oceanica coccospheres to have an "unchanged" coccosphere diameter under a past high-CO2 condition with approximately five times higher DIC levels than modern (11.3 mM DIC, $\emptyset$ = 6.50 $\mu$m compared to 2.3 mM DIC, $\emptyset$ = 6.56 $\mu$m). However, Rickaby et al. (2010) show that C. braarudii POC increases with increasing DIC. This would suggest that the effect of DIC concentration on cellular POC maybe species-specific. We inferred POC changes from measured cell size, whereas the POC to volume relationship may or may not remain constant depending on the macromolecular composition (i.e., a change in $\emptyset$ may or may not equal a change in POC). However, the possible influence of DIC on cellular POC will be added to L278.

31: L 300 - in results, max. $\mu$ for H. carteri is listed as 0.45 d-1.

Author response: $\mu$ = 0.45 d-1 as reported in the results is correct and L300 has been changed.

32: L 300-303 - This argument is not convincing. If coccosphere geometry is to be a tool for looking at long-term (as this is the fossil record) changes in growth, then it is necessary to look at reduced physiological rates due to e.g. lower nutrient availability. It is not possible to look at long-term shifts in growth phase in the fossil record. If there is an increased percentage in large cells in a certain sediment layer, what does this mean? That this was a time of "nutrient limitation" so that more cells grew larger and stopped dividing? Although this hypothesis is intriguing, it is necessary to look for a link between reduced physiological rates and larger cells if this is to be applied as a proxy to the fossil record.

Author response: Here we are expanding the concepts demonstrated in Gibbs et al. (2013) into three further species to look at the effect of reduced physiological rates due to lower nutrient availability on coccosphere geometry in the context of nutrient

depletion towards the end of a batch culture experiment, a widely employed culturing approach. As stated in response to a previous comment, future work could indeed supplement the presented dataset and analysis with data from a continuous or a semi-continuous approach and would provide valuable additional information to strengthen future investigations of growth in the fossil record of coccolithophores. This has now been added to the discussion. But we do not agree that this is the only way to examine coccosphere geometry changes with growth phase to look at long-term shifts in growth phase in the fossil record, or that this detracts from the research, analysis and interpretation presented. In answer to the reviewer's comment "If there is an increased percentage in large cells in a certain sediment layer, what does this mean?" we show in Section 4.3 L 344-345 and Figure 5 that an increased percentage of coccospheres with a CN typical of 'ready to divide' cells (i.e., greater than the 90th percentile of the CN data) would be indicative of less favourable conditions for growth, where the population has shifted towards the coccosphere geometry characteristics of a population growing non-exponentially. To attribute such a response solely to nutrient availability would be an unrealistic simplification, rather we consider it a response to a 'poor growth environment' that could include any factor or combination of factors that act to slow population growth. The modern field data presented by Gibbs et al. (2013) and shown in Fig. 5 contrasting the mean coccosphere geometries of bloom and non-bloom communities lends real strength to this approach and interpretation. This approach was also demonstrated across the PETM in O'Dea et al. (2014; their figure 1f) where the percentage of coccospheres exhibiting 'slowed division' (based on CN) remained relatively constant in Toweius throughout the interval "indicating that high levels of cell division are maintained across the PETM" whereas coccosphere geometry indicated slowed cell division in Coccolithus across the same interval. Our data supports that coccosphere geometry (CN and Ø) can be used as a qualitative indicator of changes in growth state but we do not claim to be able to quantitatively determine growth rates or other physiological rates directly from fossil coccospheres. Our approach is presently the only method that can be used to access any degree of growth information in the fossil record. We make the

relative limitations of this approach to field or fossil samples very clear - that "growth phase can be estimated" (L 355) and that "we must be clear that the environmental and growth signal recorded in field populations is always more complex than any laboratory experiment results" (L 363-364) before listing considerations that would affect the signal observed in L364-375.

33: L 310 - in this study, coccolith length is independent of growth phase so this factor of coccosphere geometry should be removed here.

Author response: To improve clarity, L310 has been re-phrased, as Ø, CL and CN are species specific, but Ø and CN respond identically to growth phase changes. It now reads: "A notable finding of this study is that coccosphere geometry (coccosphere size, coccolith length and coccoliths per cell) is species-specific but Ø and CN responds identically to growth phase across four different species of Calcidiscus, Coccolithus and Helicosphaera."

34: L 416 - Gerecht et al. (2015) show that growth rate determines calcite production in C. braarudii: PIC production is reduced by ca. 50% due to the 50% reduction in growth rate.

Author response: This reference has been added to L416.

35: L 420 - (Toweius) pertusus needs to be cursive

Author response: Corrected.

36: Table 1 - is this all combined data i.e. exponential and stationary phase? Table 1: check values Max PIC for C. braarudii (lower than Mean) Table 1: there is no mention in the methods as to how POC per cell was calculated, nor is it presented in the results or discussed in the rest of the paper.

Author response: All of the data for Table 1 has been checked against the original data and corrected where appropriate. An additional supplementary table now summarises coccosphere geometry separately for exponential and non-exponential days as well.

POC data has been removed from Table 1 as it is not discussed in the manuscript.

37: Figure 2 - check frequency values for right y-axis.

Author response: Corrected from 50 to 5.

38: Figure 3 - How is "early stationary-phase growth" defined i.e. which daily growth rate is still considered exponential?

Author response: Early stationary phase growth describes informally the day(s) immediately following the departure of a plot of cell ml-1 vs. time data from an exponential curve fitted to definitely exponential days of growth. To avoid confusion and maintain consistency, the use of early stationary-phase growth in the description of Figure 3 in the text has been changed to "non-exponential".

39: Figure 4 provides convincing evidence that coccosphere geometry data can be obtained by POL. I wonder, however, why SEM was not used? This would supply more accurate measurements of coccosphere diameter and especially of CL and equally adequate estimates of CN. It would not provide cell diameter, but the authors do not discuss cell diameter in the text. Cell diameter should be mentioned in the results as it is included in table 1, even if only as a short sentence e.g. cell diameter followed the same trend as coccosphere diameter (?).

Author response: We use LM for this study for a number of reasons. 1) It is very fast to prepare and analyse a large number of samples. The observation of coccospheres (culture, plankton or fossil) can be very time consuming using the SEM. 2) These species have morphologically 'robust' coccoliths and coccospheres, that is to say that the edges of the coccosphere (internal and external) and coccoliths (under polarised and plane light on the cross-polarised LM) are very clear. 3) CN can be broadly estimated from SEM but LM allows us to count each individual coccolith in the coccosphere and this therefore highly accurate in comparison. 3) The LM reveals both external and internal dimensions of the coccosphere, whereas the SEM would

generally only reveal the external. A sentence has been added into the results with the cell size data as suggested. It is essential for our coccosphere research that all measurements are obtained from the same individual coccosphere to investigate ratios between parameters on a cellular-level and this would not be possible with a mix of LM and SEM work.

40: The authors present calcite production rates in Figure 6. I would appreciate a sentence on how this was calculated in the Methods section i.e. was this an average of all PIC contents x growth rates in exponential phase or just from one specific day? Likewise for the "stationary phase"-values.

Author response: We state the calculation used for PIC production as Eqn. 1 in Sect. 4 and have added a sentence here to specify what data was used for which calculation. In the Fig. 6 caption we do say that (a) that the mean and 25th to 75th percentiles are calculated from exponential days or non-exponential days respectively (data from 22 °C experiment) and (b) shows the full range (minimum to maximum) in PIC, exponential and non-exponential growth rates that we observed respectively across our entire dataset to represent the absolute minimum and maximum values that could occur. Variance in the species-specific shape factor taken from Young and Ziveri (2000) is not included in the PIC calculations.

41: I have found at least 3 errors: de Vargas 2004 and Keller et al. 1987 are missing, whereas Young 2003 is in the citation list, but not cited in the text. Ziveri 2007: check journal name

Author response: de Vargas is in the reference list but under 'V' and so has been moved to 'd'. Keller et al. (1987) has been added, Young (2003) removed, Ziveri (2007) journal name has been changed to the ISI Journal Abbreviation of "Deep-Sea Res. Pt. II". All references have now been checked for accuracy.

References mentioned in responses:

Benner, I., Diner, R. E., Lefebvre, S. C., Li, D., Komada, T., Carpenter, E. J. and Stillman, J. H.: Emiliania huxleyi increases calcification but not expression of calcification-related genes in long-term exposure to elevated temperature and pCO2, Phil. Trans. R. Soc. B., 368, 20130049, doi: 10.1098/rstb.2013.0049, 2013.

Bown, P. R., Gibbs, S. J., Sheward, R. and O'Dea, S. A.: Searching for cells: the potential of fossil coccospheres in coccolithophore research, J. Nannoplankt. Res., 34, 5–21, 2014.

Collins, S., Rost, B. and Rynearson, T. A.: Evolutionary potential of marine phytoplankton under ocean acidification, Evol. Appl., 7, 140-155, doi: 10.1111/eva.12120, 2013.

Daniels, C. J., Sheward, R. M. and Poulton, A. J.: Biogeochemical implications of comparative growth rates of Emiliania huxleyi and Coccolithus species, Biogeosciences, 11, 6915–6925, doi: 10.5194/bg-11-6915-2014, 2014.

Finkel, Z. V., Beardall, J., Flynn, K. J., Quigg, A., Rees, T. A. V. and Raven, J. A.: Phytoplankton in a changing world: cell size and elemental stoichiometry, J. Plankton Res., 32, 119-137, doi: 10.1093/plankt/fbp098, 2010.

Gibbs, S. J., Poulton, A. J., Bown, P. R., Daniels, C. J., Hopkins, J., Young, J. R., Jones, H. L., Thiemann, G. J., O'Dea, S. A. and Newsam, C.: Species-specific growth response of coccolithophores to Palaeocene–Eocene environmental change, Nat. Geosci., 6, 218–222, doi:10.1038/ngeo1719, 2013.

Jin, P., Gao, K. and Beardall, J.: Evolutionary responses of a coccolithophorid Gephyrocapsa oceanica to ocean acidification, Evolution, 67, 1869-1878, doi: 10.1111/evo.12112, 2013.

Lohbeck, K. T., Riebesell, U. and Reusch, T. B. H.: Adaptive evolution of a key phytoplankton species to ocean acidification, Nat. Geosci., 5, 346-351, doi: 10.1038/NGEO1441, 2012.

Lohmann, H.: Die Gehause und Gallertblasen der Appendicularien und ihre Bedeutung fur die Erforschung des Lebens im Meer, Verhandlungen Deutsche Zoologische Geselleschaft, 19, 200-239, 1909.

Moolna, A. and Rickaby, R. E. M., Interaction of the coccolithophore Gephyrocapsa oceanica with its carbon environment: response to a recreated high-CO2 geological past, Geobiology, 10, 72-81, doi: 10.1111/j.1472-4669.2011.00308.x, 2012.

O'Brien, C. J., Peloquin, J. A., Vogt, M., Heinle, M., Gruber, N., Ajani, P., Andruleit, H., Arístegui, J., Beaufort, L., Estrada, M., Karentz, D., Kopczyńska, E., Lee, R., Poulton, A. J., Prtichard, T. and Widdicombe, C.: Global marine plankton functional type biomass distributions: coccolithophores, Earth Syst. Sci. Data, 5, 259-276, doi: 10.5194/essd-5-259-2013, 2013.

Rickaby, R. E. M., Henderiks, J. and Young, J. N.: Perturbing phytoplankton: response and isotopic fractionation with changing carbonate chemistry in two coccolithophore species, Clim. Past, 6, 771-785, doi: 10.5194/cp-6-771-2010, 2010.

Sieburth, J. McN., Smetacek, V. and Lenz, J.: Pelagic ecosystem structure: Heterotrophic compartments of the plankton and their relationship to plankton size fractions, Limnol. Oceanogr., 23, 1256-1263, doi: 10.4319/lo.1978.23.6.1256, 1978.

Šupraha, L., Gerecht, A. C., Probert, I. and Henderiks, J.: Eco-physiological adaptation shapes the response of calcifying algae to nutrient limitation, Sci. Rep., 5, 16499, doi: 10.1038/srep16499, 2015.

Young, J. R., Bergen, J. A., Bown, P. R., Burnett, J. A., Fiorentino, A., Jordan, R. W., Kleijne, A., van Niel, B. E., Romein, A. J. T. and von Salis, K.: Guidelines for coccolith and calcareous nannofossil terminology, Palaeontology, 40, 875-912, 1997.

Young, J. R. and Ziveri, P.: Calculation of coccolith volume and its use in calibration of carbonate flux estimates, Deep-Sea Res. Pt. II, 47, 1679-1700, doi: 10.1016/S0967-0645(00)00003-5, 2000.

Response to Reviewer #2

Comments:

Presents novel results, is very well written, has a well reasoned rationale and an overall well structured text, and clear figures. The study addresses an interesting topic. . . the data presented are new and of good quality and do support the conclusions drawn by the authors. In addition, the manuscript is presented in a way that will be also accessible to non-specialists, which is an added value for publication in a multidisciplinary journal such as Biogeosciences.

Specific Comments:

1: Much emphasis is put on the relevance of this study for the investigation/interpretation of coccolithophore geometries in the fossil record as a proxy for coccolithophore growth phase. However, in the introduction the authors state that intact fossil coccospheres can be found in '. . .exceptionally well-preserved sedimentary deposits. . .'. I would therefore suspect that the application of this growth phase proxy is perhaps useful only in a very limited number of settings and of a few geologic periods where/when intact fossil coccospheres are found. That being the case, the statements about the relevance of this study for the interpretation of the palaeorecords (also in the conclusions) should be toned down, at least in the terms used by the authors. Given that this is not the main reason why this is a valuable piece of work, these statements could be toned down without affecting the relevance and novelty of the study.

Authors response: The presence of coccospheres in sediments is indeed atypical, and our reference to '...exceptionally well-preserved sedimentary deposits...' refers to the very high preservation quality of the calcareous nannoplankton seen in the sediments used by Gibbs et al. (2013) for their study. In other research by the authors (Sheward et al., in prep), $\sim$20 to >300 coccospheres per standard smear slide (see Gibbs et al., 2013 and Bown et al., 2014) have been documented and measured for over 4,000 individual coccospheres of Paleogene age from 11 different sites that range in paleo-latitude between $\sim$58 °S and $\sim$47 °N and from the North Atlantic Ocean, North

[Figure]

Pacific Ocean, Indian Ocean and Southern Ocean. Hence, whilst uncommon in large numbers, there is certainly a large range of Paleogene-age sites where intact coccospheres have been found in numbers suitable for robust quantitative analysis. This is likely to be true of other geological periods and sites, but reasoned selection of sections for sampling will be important as hemipelagic sediments (particularly those with less intense bioturbation) are perhaps more likely to contain coccospheres than deep-sea oozes. Several other publications report observations of coccospheres from a variety of locations and ages (e.g., Covington, 1985; Mai et al., 1997; Mai et al., 1998; Mai, 1999). The approach that we present for the interpretation of coccosphere geometry can also be readily applied to coccospheres found in modern-day plankton samples (see Gibbs et al., 2013), or in surface or Holocene sediments. As these points are not presently expressed in the paper, a statement about applicability is now added into Section 4.3 including describing the present occurrence of fossil coccospheres and we re-worded L 432 – 435 to better state this in the conclusions.

2: Often times in the manuscript it is stated that results are statistically significant, but a section in the methods that specifically presents the statistical approaches used in this study is missing and should be added. Also, data analysis could benefit from some (bootstrap?) outlier analysis, specifically when different properties of the coccolithophore geometry are regressed against one another (e.g., Fig. 2e-h). This would certainly improve the analysis of the high quality (and rich) dataset presented in this study.

Authors response: We agree with Reviewer 2 that a specific methods section presenting our statistical approaches would be beneficial. This methods section now states as follows: "The relationship of Ø with CL and CN in each species were tested by Model II reduced major axis (RMA) linear regression analysis and confidence intervals (95 %) for the regression slope were calculated by bootstrapping over 1999 iterations using the freeware Paleontological Statistics (PAST; v. 3.13; Hammer et al., 2001). We compare species-specific mean Ø and mean CN between growth phases using a t-test

in GraphPad Prism (version 7.0a for Mac OS X; GraphPad Software, Inc., USA). The difference in mean Ø or CN between exponential-phase growth and non-exponential-phase growth were considered significant at $p < 0.05$."

References mentioned in responses:

Bown, P. R., Gibbs, S. J., Sheward, R. and O'Dea, S. A.: Searching for cells: the potential of fossil coccospheres in coccolithophore research, J. Nannoplankt. Res., 34, 5–21, 2014.

Covington, M.: New morphologic information on Cretaceous nannofossils from the Niobrara Formation (Upper Cretaceous) of Kansas, Geology, 13, 683-686, doi: 10.1130/0091-7613(1985)13<683:NMIOCN>2.0.CO;2, 1985.

Gibbs, S. J., Poulton, A. J., Bown, P. R., Daniels, C. J., Hopkins, J., Young, J. R., Jones, H. L., Thiemann, G. J., O'Dea, S. A. and Newsam, C.: Species-specific growth response of coccolithophores to Palaeocene–Eocene environmental change, Nat. Geosci., 6, 218–222, doi:10.1038/ngeo1719, 2013.

Hammer, Ø, Harper, D. A. T. and Ryan, P. D.: PAST: Paleontological statistics software package for education and data analysis, Palaeontologia Electronica, 4, 9pp., 2001.

Mai, H.: Paleocene coccoliths and coccospheres in deposits of the Maastrichtian stage at the 'type locality' and type area in SE Limburg, The Netherlands, Mar. Micropaleontol., 36, 1-12, doi: 10.1016/S0377-8398(98)00023-1, 1999.

Mai, H., Perch-Nielsen, K., Willems, H. and Romein, T.: Fossil coccospheres from the K/T boundary section from Geulhemmerberg, The Netherlands, Micropaleontology, 43, 281-302, 1997.

Mai, H., Hildebrand-Habel, T., Perch-Nielsen, K. and Willems, H.: Paleocene coccospheres from DSDP Leg 39, Site 356, São Paulo Plateau, S Atlantic Ocean, J. Nannoplankt. Res., 20, 21-29, 1998.

Please also note the supplement to this comment:
http://www.biogeosciences-discuss.net/bg-2016-435/bg-2016-435-AC1-
supplement.pdf